# Unpacking musical beauty: Sound, emotion, and impact differences across expertise and personality

Yuko Arthurs[1,2]*, Eve Merlini[1,3], Diana Omigie[1]

**1** Department of Psychology, Goldsmiths, University of London, London, United Kingdom, **2** Department of Psychology, Waseda University, Tokyo, Japan, **3** School of Psychology, Cardiff University, Cardiff, United Kingdom

* y.arthurs@gold.ac.uk

## Abstract

Beauty is an aesthetic quality that many composers and performers strive to attain and that most listeners value highly in music. Yet, what listeners hear and feel when they find music beautiful, the impact this experience has on them, and wherein lie any potential individual differences remain unclear. To address these gaps, an online questionnaire was administered to 81 adults. They described their experience of listening to three self-selected "beautiful pieces of music" alongside three pieces they listened to a lot but did not consider beautiful; focusing on each piece's sonic features, emotions induced, and any impact engaging with the pieces had had on them. Respondents' personality traits and level of musical training were also collected to explore if these contributed significantly to variability in the data. Thematic analysis of text responses showed that participants consider intrinsic features of the music (mellow timbre, pleasing melodies and harmonies, slow tempo), cognitive factors (degree of change, complexity levels, and repetition), and aesthetic criteria (emotional effects, presence of a message, expressivity) when judging the beauty of a piece. It also showed that, beyond being calming and evoking pleasant sadness, pieces perceived as beautiful tend to move, transport and emotionally resonate with the listener, inspiring musical activities, and providing strong emotional support. Exploratory, quantitative analyses revealed that amateur musicians and open individuals reported a wider range of contributing musical features, while both amateur and professional musicians, along with conscientious individuals, reported a broader range of musical beauty's influence on their musical activities. Our study suggests that musical beauty is a multifaceted phenomenon that has profound emotional impact on most individuals, while also displaying notable sensitivity to listeners' personality and musical background. In doing so, our study contributes to a more comprehensive account of what it means to experience beauty in music.

**Data availability statement:** All data used for the purposes of this study are available from the following OSF link: https://osf.io/nqph4*****A/PROS AT ACCEPT: Please follow up with the authors for OSF repository information available at Accept.***** All data files will be available from OSF website after acceptance.

**Funding:** This work was supported by Grant-in-Aid for Japan Society for the Promotion of Science Fellows awarded to YA (No. 22J40130, JSPS, https://www.jsps.go.jp/english/).The funders had no role in study design, data collection and analysis, decision to publish, or preparation of the manuscript.

**Competing interests:** The authors have declared that no competing interests exist.

## Introduction

Beauty is a valued quality and core concept in the aesthetic experience of music [1–4]. Indeed, it has been noted variously that many composers strive to attain beauty [5,6]; musicians often aim to achieve performances that audiences might find beautiful and expressive [7]; and people regularly choose to listen to music to savour its beauty [8,9]. However, despite its importance, research on musical beauty from a psychological perspective is still relatively limited [10–13].

Experiencing beauty is considered a relatively common occurrence [14] that can emerge from encountering artworks, nature [14,15], and human morality [16,17]. Nevertheless, a question for many scholars is what leads people to arrive at beauty judgements. Indeed, while certain features have been found to be predictive of beauty judgements across a range of domains (e.g., degree of symmetry), others (e.g., hue, saturation of colours used) are much less relevant outside a narrow set of experiences. Perhaps, at least partly for such practical reasons, and despite its ubiquitousness, beauty has to date been considered difficult to define [18–20].

Also frequently debated is how beauty should be conceptualised for empirical study. Beauty has been considered by some psychologists as a judgement towards an object [21,22], at the same time as it has been described as a particular, highly valued, feeling or emotion that an observer may experience when encountering an object [15,23]. Yet, other contemporary psychologists have operationalised beauty as being one of several possible criteria for arriving at a positive aesthetic judgement [3,9,24], where recognising beauty is generally associated with more positive aesthetic judgements.

An aesthetic judgement is considered an output, along with emotion and preferences, of the processing of artworks and other objects. This form of processing is held to involve perceptual, cognitive, and affective components [21,25,26]. Some argue, though, that it also has consequences beyond the observer's emotions, such as influencing behaviours or attitudes [27]. Interestingly, individual differences, such as expertise in the field and personality traits, are increasingly hypothesised to influence experiences of beauty [3]. Yet, there has been little empirical research addressing the question and the limited findings so far remain inconclusive [14].

Recent studies have used wide-ranging methods to tackle some of the fundamental questions regarding musical beauty. One study by Fleckenstein et al. [12] was particularly valuable in throwing light on participants' subjective definition of musical beauty. Their study revealed that participants mainly view musical beauty as both an emotional experience on the one hand and as determined by the properties of an object on the other. The study also identified various factors that contribute to the experience of musical beauty, including participants' felt emotions, the emotions expressed by the music, the timbre of instruments, and the memories listeners associated with the music. Complementing the qualitative approach taken in that study, Brattico et al. [10] investigated the neural correlates of the experience of musical beauty and implicated the orbitofrontal activity and bilateral supratemporal activity. That study also revealed differences in sonic features of beautiful versus not beautiful

sections of music; beautiful, compared to not beautiful, sections were rated (by composers) as more tonal, simple, melodic, traditional, calm and sad.

However, while these recent studies [10,12] confirm that musical beauty is associated with certain sonic features and rich emotional content, there are still notable gaps in the literature. For instance, while Brattico and colleagues [10] compared sonic features of beautiful and not beautiful sections of three pieces of music, direct comparisons of *beautiful* with *not beautiful* pieces of music—as defined by the listener— are still rare. Moreover, although Fleckenstein et al.'s study [12] tentatively indicates that musical beauty may have meaningful short-term impact on the listener—such as losing one's sense of self, altered consciousness and being immersed, the long-term impact of experiencing musical beauty remains largely unexplored. Lastly, the extent to which the experience of musical beauty varies according to individual differences has not, to our knowledge, been examined, unlike in visual arts, where such questions have received significant attention [28,29].

These considerations thus raise several questions that we argue could benefit from further research, namely: What musical features do listeners think make a piece of music beautiful? What do they feel when they find music beautiful versus when they do not? What is the impact of experiencing musical beauty? And, finally, one could also ask: To what extent is the listener's experience of musical beauty influenced by their musical expertise and personality traits?

## Features of music perceived as beautiful

The extent to which specific sonic features might be tied to key experiences of music (such as chills, being moved and the feeling of awe) has seen significant interest in the music psychology literature [30–33]. These studies have aligned in highlighting the role of both lower-level musical elements (such as loudness, tempo, and timbre) and higher-order, structural characteristics (such as tonal clarity) in affording these experiences. Regarding musical beauty, consonance—two or more harmoniously sounding concurrent tones—has been traditionally associated with beauty in the context of Western tonal music [34,35]. More recently, Fleckenstein et al.'s [12] participants reported not just harmonic content, but also instrumental timbre and tempo/rhythm/meter as factors contributing to their experience of musical beauty.

Interestingly, studies which have examined sections of pieces considered beautiful by listeners highlight the potential importance of temporal changes in musical elements for the experience of musical beauty. Beautiful compared to not beautiful sections in music have been associated with more tonal, simple, melodic, traditional features [10], raising the possibility that listeners may find these moments more beautiful due to how they contrast with other moments. Similarly, a study by Omigie et al. [13] demonstrated that beautiful passages or sections of music are often characterised by change, such as a drop in tempo and a reduction in polyphony in some cases, or by increases in dynamics, pitch register, major tonality, and harmonic clarity, in others.

These findings, showing the potential importance of change in the context of musical beauty, align with an argument that when preceded by dissonance the beauty of consonance may be enhanced due to the contrast between them [36,37]. On a more abstract level, it has been suggested that the structure of music—i.e., the way in which musical elements are arranged to form a piece—and the cognitive processes such structure entails may be a key contributor to the experience of beauty. On the one hand, simplicity might be expected to promote experiences of beauty. Indeed, according to the processing fluency theory [38], perceived beauty may be influenced by the ease with which a piece can be cognitively processed, where the easier it is to process the piece, the more beauty may be experienced. On the other hand, beauty judgements are likely also intrinsically related to the listener's expectations, which are based on their knowledge of musical syntax [37,39,40]. Here, it is recognised that while it may be a positive experience for the listener when a piece of music unfolds as anticipated [40,41], a degree of unexpectedness, surprise, or novelty in music can lead to greater feelings of pleasure [42,43], an experience that (as we discuss below) people associate with beauty.

Taken together, the mechanisms underlying musical beauty judgements would appear to be open to scrutiny. Here, an open question that remains is whether the features that contribute to beauty experiences are limited to simplicity and ease of processing, or extend to more complex cognitive processing.

## Feelings when experiencing beauty in music

What do listeners feel when they experience musical beauty? In a large-scale study by Brielmann et al. [15], participants reported that when they experience beauty from images, music, or reflect on personal memories, they feel that their beauty experience is universal and shared with others. Participants also reported feeling a strong desire to continue the experience, in line with claims from philosophers like Plato and Kant [15]. Brielmann et al. [15] found that for many people, beauty is closely related to pleasure, a link that dates back to the Ancient Greeks [44], and which several empirical studies have consistently shown [13,45]. Indeed, Skov and Nadal [46] go so far as to equate beauty with pleasure, arguing that the perception of beauty seems to arise from the neurobiological system in the same way that hedonic evaluations (such as liking and pleasure) do.

Many authors, however, have sought to examine the range of emotions that may co-occur with or characterise the experience of beauty [11–13,15], based on the view that emotions are an important part of the aesthetic experience [16,23,47,48]. This research shows that, alongside positive valence or pleasure, experiencing beauty from artworks is often linked with the feeling of being moved [33,49,50], the feeling of positive awe [32], and the feeling of sadness [51–54]. Further, hinting at the additional role that epistemic emotions may play, Armstrong and Detweiler-Badell [47] argue that the desire to understand a piece and the satisfactory resolution of such desire through apprehending the piece, are essential for experiencing beauty.

The variety of emotions associated with beauty experiences so far suggest that characterising beauty as pleasure alone may not fully capture its multifacetedness, compared to other forms of pleasure [46]. However, in light of ongoing debates about the appropriateness of distinguishing aesthetic pleasure from other forms of pleasure [46], systematic research into the emotions that accompany beauty experiences may be considered increasingly necessary.

## The impact of experiencing beauty in music

Finally, highly worthy of exploration is whether certain experiences of music have an impact on behaviours, actions and perception of self, including in connection with others [27,55–57]. One possibility is people will tend to want to repeat the listening experience of beautiful music, embedding it into their lives in particular and potentially strategic ways, both because of the pleasure [13,15] and the other emotions that beauty tends to afford (e.g., pleasant sadness). Indeed, it is widely accepted that people seek out experiences they know they like or find pleasurable, thanks to the behaviour-reinforcing nature of the human reward system [58,59]. It is also recognised that they widely seek out emotional stimuli for the sake of emotion regulation and catharsis [60–63].

Yet, another possible impact of beauty in music is motivating further engagement with music including in ways that require effort and commitment, such as learning an instrument or joining a music group. It has been reported that people who pursue music professionally often had strong emotional and aesthetic experiences of music during childhood [64,65], or a strong sense of calling for music during adolescence [66]. It thus seems likely that experiencing beauty in music may be a profound trigger for individuals to make a lifelong commitment to music engagement.

Last but not least, beyond musical engagement, another possible impact of experiencing musical beauty is inducing feelings of self-transformation. Previous studies show that engagement with the arts can sometimes lead to individuals having an increased sense of purpose in their lives [56], stronger emotional resilience [56], deeper connection with self and others [56,67], more empathetic attitudes towards others [67–69], and greater awareness of different cultures and values [70,71]. However, while it seems plausible that these shorter- and longer-term impacts may emerge from experiences of musical beauty specifically, this has not been examined in the extant literature on musical beauty.

## Individual differences: Musical training and personality

Finally, even though philosophical notions of the universality of the beauty experience have seen some support, e.g., [15,44], it is widely accepted that beauty judgements can be highly subjective [72–74]. Expertise is considered an important individual difference in the context of music experiences [75]. It has been reported that musicians and those who

engage with musical activities, compared to nonmusicians, tend to be more sensitive to beauty [76], experience more reward from music [77], experience chills from music more frequently [78,79], and have finer and more intense emotional and aesthetic responses [75,80,81].

Musical training, specifically, improves one's ability to process both acoustic [82,83] and musical elements [84–87]. Importantly, all of these may, in turn, be expected to drive differences in how beauty is experienced since perceiving and cognitively processing the features of a piece is considered an integral part of the aesthetic experience of music [21,25,26]. Beyond perceptual abilities, musical training may also alter the aspects of a piece that a listener relies on when making evaluative judgements [88–90]. A study by Spitzer and Coutinho [88] found that while nonmusicians tended to make judgements based on lower-level sonic features of a piece when trying to determine the emotional character of a piece, musicians tended to consider higher-level, structural aspects alongside such low-level, sonic features. Similarly, an ERP study by Müller et al. [89] demonstrated that nonmusicians, compared to musicians, rely more on their emotions when making aesthetic judgements of music.

These differences between musicians and nonmusicians may or may not result in differing experiences of beauty; both in terms of features that contribute to the judgements of beauty, and to the feelings that are most salient in response to musical beauty. Further, since it has been reported that people who pursue music professionally often had strong emotional and aesthetic experiences of music during childhood [64,65], or a strong calling for music during adolescence [66], one could predict that musicians, more than non-musicians, will report having been more impacted by musical beauty over the course of their lives. Here, it is important to note that being a musician may provide more opportunities to experience the possible impacts of music, and that as such impacts reported by this group may inevitably be higher. In either case, a positive relationship between musicianship and breadth or scale of impacts from beautiful music seems a reasonable prediction to make.

Finally, personality traits, commonly modelled as five dimensions (Extraversion, Neuroticism, Conscientiousness, Agreeableness, and Openness to Experience), are yet another factor commonly thought to influence an individual's aesthetic and emotional experience of music [74,91]. Among these, openness to experience has been shown to be especially important, being associated with a tendency for greater aesthetic experiences such as chills and awe from music [74,79], with seeking out aesthetic experience [92] and intellectual stimulation [93] from listening to music, and with greater enjoyment and experiencing more positive and fewer negative emotions from music [91,94,95]. Evidence also links this trait to enhanced musical cognitive skills—specifically, more fine-grained abilities to discriminate melodic and rhythmic patterns [96]. Finally, openness to experience has been shown to moderate well documented effects of exposure on people's aesthetic evaluations [97], influencing the extent to which a listener will show a tendency to value objects more highly [98,99], or conversely show reductions in liking and interest [100–102] with increasing exposure to the object.

While less systematically, other dimensions have also been shown to influence individuals' aesthetic and emotional experience of music. For example, it has been reported that extravert people tend to experience positive emotions from music [52,54,91] more intensely [91]; that people high in conscientiousness are less likely to experience negative emotions such as sadness, anger, or anxiety from music [91]; and that listeners high in neuroticism tend to experience more nostalgia [103,104] as well as negative emotions from music [91,94].

Taken together, the findings from previous studies point to the possibility that musical beauty may subjectively differ according to the listener's musical training and personality traits. Musicianship and openness to experience may influence the ability to discuss features contributing to their experience of musical beauty, while openness to experience and/or extraversion may lead to more varied experience of emotions in response to music perceived as beautiful. Musicians may also be expected to report a wider range of impacts from music.

## The present study

Inspired by the literature described above and motivated by gaps therein, this study explored how the experience of musical beauty can be understood in terms of musical features, listeners' experienced emotions, and the impact of

experiencing musical beauty, including its potential short- and long-term effects. We also sought to explore whether individual differences in the experience of musical beauty may be explained by musical training and personality traits. Uncovering distinctive features, emotions, and impacts of the experience of musical beauty promises a better understanding of what shapes the experience, the extent to which it is commonly shared or subjective, and its uniqueness or otherwise. Moreover, a fuller account of the experience of musical beauty promises a better understanding what motivates people to engage with beautiful music and artistic activities more generally.

With this objective in mind, we used an online questionnaire with open-ended questions to gather people's experiences of finding pieces beautiful or not beautiful, alongside their musical background [105,106], and personality traits [107]. Comparing the experience of beauty with its absence promised clarification of what uniquely shapes the experience of musical beauty as compared with other valued but not beauty-rich experiences of music.

Critically, concerning study design choices, we chose to employ a qualitative method for data collection and analysis due to the advantages it offers: Specifically, regarding exploring the features of sound in music described as beautiful, text responses reveal listeners' psychological impressions of both lower- and higher-level features of sound in ways that might not be possible using automated feature extraction techniques. As for emotion and impact, text responses allow the reporting of rich and diverse responses that are not obtainable if participants are only provided with pre-defined emotion labels and statements [108,109]. Similarly, while advances in natural language processing are making such methods a powerful option for text analysis in psychology, qualitative thematic analysis was opted for here to ensure that a deep and nuanced understanding of the data could be achieved. However, recognising the gap in the literature on how individual differences influence the beauty experience, our analysis strategy also involved an exploratory quantitative interrogation of the extent to which the breadth of content encapsulated in participants' responses was influenced by their personality and musical expertise. In this way, we were able to extend previous studies that have examined the beauty experience with only little concern for the role of individual factors.

## Methods

### Ethics statement

Ethics approval was obtained from the Ethics Committee of the Department of Psychology at Goldsmiths, University of London. Participants gave online written consent prior to participating in the questionnaire and received monetary compensation for their time.

### Participants

A total of 81 participants (female: $n = 42$, male: $n = 36$, non-binary/third gender: $n = 2$, prefer not to say: $n = 1$; $M_{age} = 36.30$, $SD_{age} = 15.73$, range 18–72 years old) voluntarily took part in the survey, having been recruited through the online participant recruitment platform, Prolific and through social networks between 12 January and 13 March 2024. One eligibility criterion for participation was to like listening to music. All participants were residents of the UK; however, 19.75% of the participants ($n = 16$) were originally from other countries, including Italy: $n = 4$, Nigeria: $n = 4$, the USA: $n = 2$, Denmark: $n = 1$, Hong Kong: $n = 1$, Lithuania: $n = 1$, New Zealand: $n = 1$, Poland: $n = 1$, and Sri Lanka: $n = 1$. Level of musical expertise was measured using the single item "Which title best describes you?" from The Ollen Musical Sophistication Index [105] following Zhang & Schubert's recommendation [106]. According to this index, our participants were: nonmusicians = 2, music-loving non-musicians = 26 (who were both assigned as "low" for the quantitative analysis), amateur musicians = 12, serious amateur musicians = 17 (who were both assigned as "medium"), semi-professional musicians = 13, and professional musicians = 11 (who were both assigned as "high"), in line with our efforts to sample participants across the range of expertise. 75% of participants (61 out of 81) had received formal musical lessons on various instruments (e.g., piano, violin, cello, oboe, saxophone, trumpet, guitar, drums, singing) for an average of 13.90 years ($SD_{years} = 5.97$), starting at an average age of 6.77 ($SD_{age} = 2.45$), whereas the other 25% of participants (20 out of 81) reported never having received formal musical lessons.

**Procedure**

An online questionnaire, consisting of three sections was implemented in Qualtrics [110]. The first section asked for participants' demographic information and musical background while the second section comprised two blocks with open-ended questions. One of these blocks inquired after pieces of music participants found beautiful and the other block asked for pieces of music they found not beautiful but listened to a lot: "*Could you tell us the details of three pieces of music which you find to be very beautiful?*" for the beautiful pieces block, and "*Now, could you tell us the details of three pieces of music you listen to often but which you wouldn't describe as beautiful?*" for the not beautiful block. Participants were asked about pieces of music they do not find beautiful to allow us to compare the experience of beauty with that of not experiencing beauty. We specified "listen to often' when referring to not beautiful music to ensure that participants did not conflate 'not beautiful' with 'disliked' or 'ugly' music they would normally try to avoid, but rather considered not beautiful music that they nevertheless engage with in everyday life. In the beautiful pieces and not beautiful pieces blocks, participants were asked to provide the title, artists/composers, and YouTube or Spotify links of three example pieces.

For each block, participants were then asked three main questions. The first question inquired after the musical features of the (beautiful and not beautiful) pieces that participants believed contributed to their experience of beauty or lack of it (hereafter, *Feature question*). The second question asked about the feelings and emotions participants experienced while listening to the specified beautiful and not beautiful pieces (hereafter, *Affect question*). The third asked participants about any impact the specified beautiful and not beautiful pieces had had on them (hereafter, *Impact question*). For the Feature and Affect questions, participants were asked to provide answers for each of the three beautiful and not beautiful pieces of music they had listed and/or for beautiful or not beautiful music in general, as relevant. For the Impact question, participants were asked to indicate whether any of the pieces had impacted them and if so to describe the nature of that impact. In the event that they couldn't report being impacted by the listed pieces, they were invited to describe the impact of any other beautiful/not beautiful pieces they had not listed. Although this study aims to explore both short- and long-term impacts of experiencing musical beauty, participants were not instructed to report impacts as short- or long-term since it is difficult to specify these systematically. The questions were phrased as follows in each block:

**Beautiful pieces block.**

1. Feature: *What do you think makes these pieces of music so beautiful, in general or for each piece separately as appropriate? Please answer in terms of the sonic features of the music (i.e., in terms of how it sounds) and/or how it is composed.*

2. Affect: *How do you feel when you listen to these beautiful pieces of music, in general or for each piece separately as appropriate?*

3. Impact: *Have any of these pieces impacted you in any way? If 'Yes', could you briefly explain? If not these pieces, have any other pieces you find beautiful had a significant impact on you? How so?*

**Not beautiful pieces block.**

1. Feature: *Why would you not describe these pieces of music as beautiful? Please answer in terms of the sonic features of the music (i.e., in terms of how it sounds) and/or how it is composed. Please answer in general or for each piece separately as appropriate.*

2. Affect: *How do you feel when you listen to these pieces of music you don't describe as beautiful, in general or for each piece separately as appropriate?*

3. Impact: *Have any of these pieces you don't describe as beautiful impacted you? If 'Yes', could you briefly explain? If not these pieces, have any other pieces you listen to often but don't find beautiful had a significant impact on you? If so, how?*

The third section of the questionnaire was a Ten Item Personality Inventory (TIPI) by Gosling et al. [107] to measure participants' personality traits. Additionally, participants' levels of engagement with artistic beauty, beauty in nature, and beauty in human morality were measured using an Engagement with Beauty Scale (EBS) by Diessner et al. [16], although the EBS data was not included in the current analysis.

### Analysis

**Qualitative analysis of open-ended text responses.** Text responses to three questions (Feature, Affect, and Impact) for both beautiful and not beautiful pieces were analysed using thematic analysis [111]. Thematic analysis was considered the most suitable approach for analysing text responses about people's lived and rich experience of beauty in music as it allowed the inductive identification of underlying themes, and because previous studies have successfully implemented this approach in examinations of people's experiences and views related to musical activities or aesthetic responses, e.g., [32,112].

We analysed text responses to questions related to the Feature, Affect, and Impact questions separately, using a codebook approach [113]. First, the researchers independently read the data several times to become familiar with it. Next, the first author coded 80% of all responses, and created codes and a codebook that all authors then evaluated for appropriateness. Finally, the second author independently coded all responses following the codebook.

Cohen's Kappa and prevalence-adjusted bias-adjusted Kappa (PABAK) was estimated per code (for the 80% of data coded by two authors) to assess inter-coder reliability and to identify any codes that might need refinement. PABAK was calculated along with Cohen's Kappa since PABAK addresses the limitation that Cohen's Kappa does not account for prevalence of codes [114]. PABAK was highly relevant in our dataset, with 8 out of 15 codes for the Feature question and 13 of 16 codes for both the Affect and Impact questions exhibiting an occurrence rate below 10% (See S1 File). Cohen's Kappa and PABAK values across codes (mean [range]) indicated moderate to very strong agreement: For Feature, Kappa = 0.78 [0.61–0.94], PABAK = 0.83 [0.55–0.97]; For Affect, Kappa = 0.85 [0.84–0.97], PABAK = 0. 87 [0.00–0.98]; For Impact, Kappa = 0.52 [0.13–1.00], PABAK = 0.90 [0.72–1.00].

**Quantitative analysis of the influence of musical training and personality traits on beauty responses.** To capture the influence of musical training and personality traits on individual experiences of musical beauty, a novel and exploratory approach was taken. Specifically, for each question, we modelled how musicianship level and personality scores predicted the number of codes present in participants' responses as they described their experience of musical beauty. The assumption was that the number of codes reflect the breadth of aspects involved in each participant's experience. To give an example, musicians might be expected to report a wider breadth (and therefore have higher count) of musical features and impacts, just as participants high in openness to experience and extraversion might be expected to report wider-ranging (and therefore have a higher count of) affective responses.

To prepare the data for such modelling, we considered participants' responses for each question separately. First, for each participant, for each text response, we determined whether a given code was relevant to that text response. We collapsed across responses given by that participant (up to 4 possible responses, given that they could report on up to three pieces and/or generally) and assigned the participant a count of one for that code regardless of how many times it was mentioned. Finally, to get our dependent variable (DV), we summed up the number of codes each participant had from each theme for each text response.

Our primary concern was on the level of breadth with which participants responded to each question, with an interest in any differences that may emerge on the theme level. Thus, two main Poisson mixed effects models were conducted on participants' responses to each question. The first, simpler, model had number of codes as the DV and included musicianship level (high, medium, low) and scores on the five personality variables as fixed effects, with participant ID included as a random effect. The second, more complex model was the same but additionally included theme as a fixed effect that could interact with each of the (individual difference) fixed effects in the first model. Poisson mixed effects models were

chosen due to the data being count data. All analyses were performed in R [115], using the *glmer* function from the lme4 package [116].

## Results and findings

### Pieces volunteered by participants

Participants listed 242 and 235 pieces of music as beautiful or not beautiful respectively. The pieces could be classified into 15 genres: 14 genres proposed by Rentfrow and Gosling [117] (alternative, blues, classical, country, electronica/dance, folk, heavy metal, rap/hip-hop, jazz, pop, religious, rock, soul/funk, and sound tracks), and pieces that were not classified into any of these 14 (such as new age, easy listening, or world music) were categorised as other. A full list of beautiful and not beautiful pieces and a description of the classification process are given in S1 Appendix.

Fig 1 shows the distribution and percentage of genres for beautiful and not beautiful pieces. As can be seen from Fig 1, nearly one third of beautiful pieces were classical pieces, followed by pop and rock, with these three genres together accounting for nearly 60% of all beautiful pieces. Similarly, 64.25% of not beautiful pieces came from these three genres, although rock accounted for the majority followed by pop and classical. Some genres, such as alternative, classical, folk,

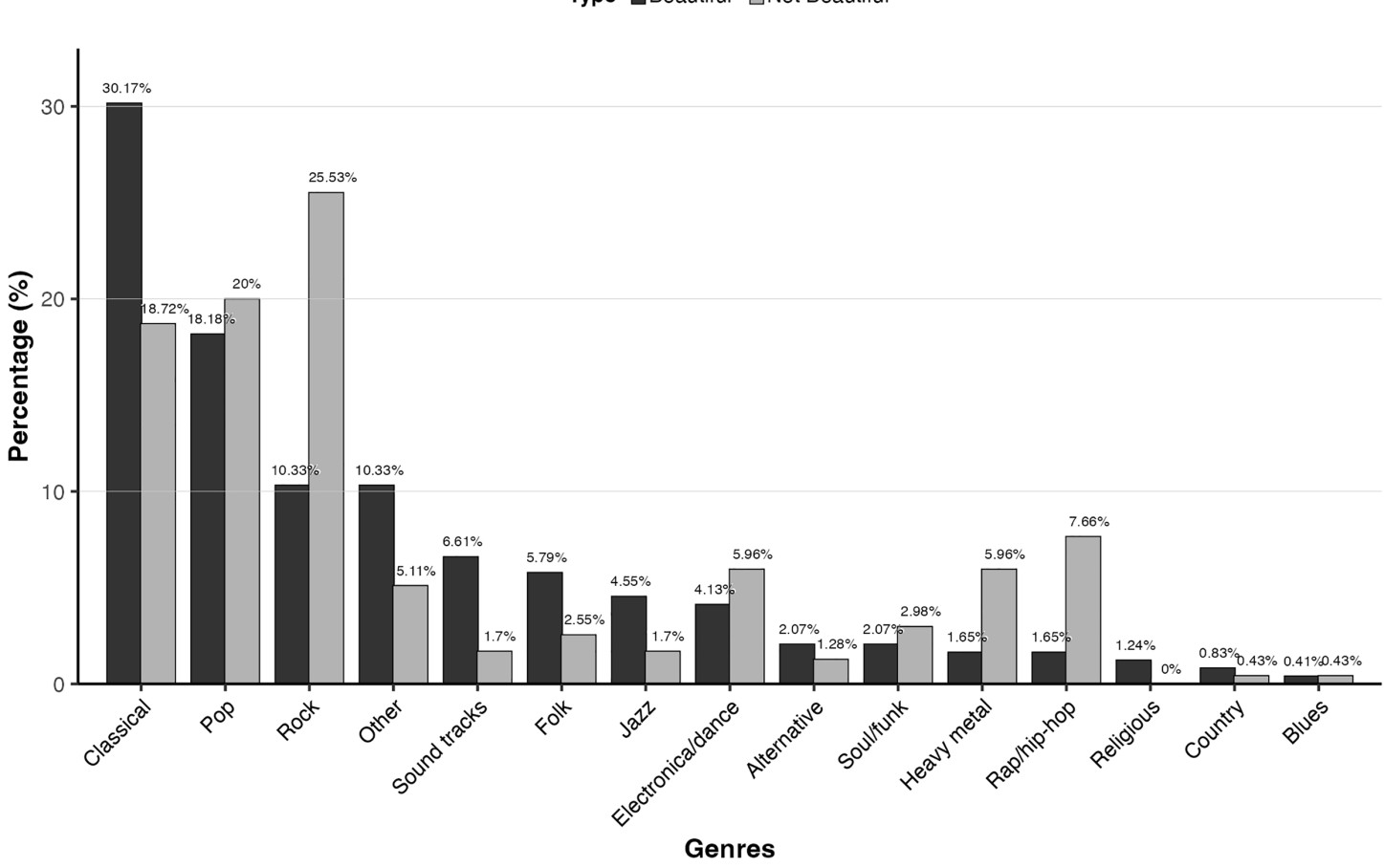

**Fig 1. Percentages of genres for beautiful and not beautiful music.** Genres were presented in the order of highest to lowest percentages of genres of beautiful music (darker bars) from left to right.

jazz, religious, sound tracks, and other, had a higher percentage of beautiful than not beautiful pieces, while electronica/dance, heavy metal, pop, rock, rap/hip-hop, and soul/funk were mentioned more often as not beautiful than as beautiful pieces. Pieces of blues, country, and religious music were the least often chosen as examples of beautiful or not beautiful music.

### Findings from thematic analysis: Feature

Analysis of text responses to the Feature question identified 15 codes, from which three themes were derived (Figs 2 and 3). Fig 2 shows the relative percentages of responses coming from beautiful and not beautiful music for each code. Fig 3 shows, for beautiful and not beautiful music separately, the number of times a given code was assigned relative to the total number of times all codes were assigned. Examples of quotes for each code in each theme, both from questions regarding beautiful and not beautiful pieces, are presented in Table 1. Quotes are shown in the text with Quote IDs in brackets, corresponding to the Quote IDs in the tables (e.g., F-"Building Blocks: Timbre"-B1). The first letter denotes the question type (F = Feature, A = Affect, I = Impact), followed by a code name, with "B" and "NB" referring to responses to beautiful and not beautiful pieces, respectively.

**Intrinsic features of sound and music.** Text responses from our participants revealed that whether or not a piece of music is perceived as beautiful is at least partially attributable to its particular physical sonic features. Commonly mentioned features included the specific timbre (F-"Building Blocks: Timbre"-B1, F-"Building Blocks: Timbre"-NB1), the quality of its melodies (F-"Building Blocks: Melody"-B2, F-"Building Blocks: Melody"-NB2) and harmonies (F-"Building Blocks: Harmony"-B3, F-"Building Blocks: Harmony"-NB3), and its slow or fast tempo (F-"Building Blocks: Rhythm and Tempo"-B4, F-"Building Blocks: Rhythm and Tempo"-NB4). Even though mentioned less frequently, the pieces' softer sounds (F-"Building Blocks: Loudness"-B5, F-"Building Blocks: Loudness"-NB5), and pitch patterns were described as affecting perceived beauty (F-"Building Blocks: Pitch"-B6). Participants also mentioned the way the piece is composed (F-"Composition Style"-B7, F-"Composition Style"-NB7), and the piece's genre (F-"Composition Style"-B8, F-"Composition Style"-NB8) as contributing factors.

**Cognitive structural factors.** Although less frequently than intrinsic features (*Building blocks* and *Composition style*), participants also referred to structural features of music—*Changes and contrasts*, *Complexity level*, and *Repetition*—as needing to be at optimal levels for a piece to be beautiful. Specifically, they mentioned changes in musical features such as timbre, loudness, tempo, and harmony as key to musical beauty (F-"Changes and Contrasts"-B9). However, not enough change was perceived as too simple (F-"Changes and Contrasts"-NB9a), while too much change was perceived as restless (F-"Changes and Contrasts"-NB9b). In line with this, participants also noted that they felt beautiful pieces have the right level of complexity (F-"Complexity"-B10), with pieces considered either too simple (F-"Complexity"-NB10a) or complicated (F-"Complexity"-NB10b) perceived as not beautiful. Finally, repetition of melodies, rhythmic patterns, or chords was described as providing a sense of structure and coherence that was characteristic of beauty (F-"Repetitions"-B11), although excessive repetition was seen as lacking variety and becoming boring, and therefore not beautiful (F-"Repetitions"-NB11).

**Aesthetic criteria.** Participants referred, with moderate frequency, to factors we named *Emotional effects, Story and message, Expressivity, Originality*, and *Performance quality* when describing what makes music beautiful or not. With regard to *Emotional effects* (and despite the question being about the *sound* of beautiful music), many participants' responses stressed that they find certain pieces beautiful because of the way the music makes them feel (F-"Emotional Effects"-B12), where pieces they did not find beautiful tended to lack the ability to move them (F-"Emotional Effects"-NB12). Similarly, the story and message transmitted by the lyrical content (F-"Story and Messages"-B13, F-"Story and Messages"-NB13), consequent feeling (F-"Story and Messages"-B14, F-"Story and Messages"-NB14), and how well the lyrical content and sound matched each other (F-"Story and Messages"-B15, F-"Story and Messages"-NB15) were described as influential to beauty experiences.

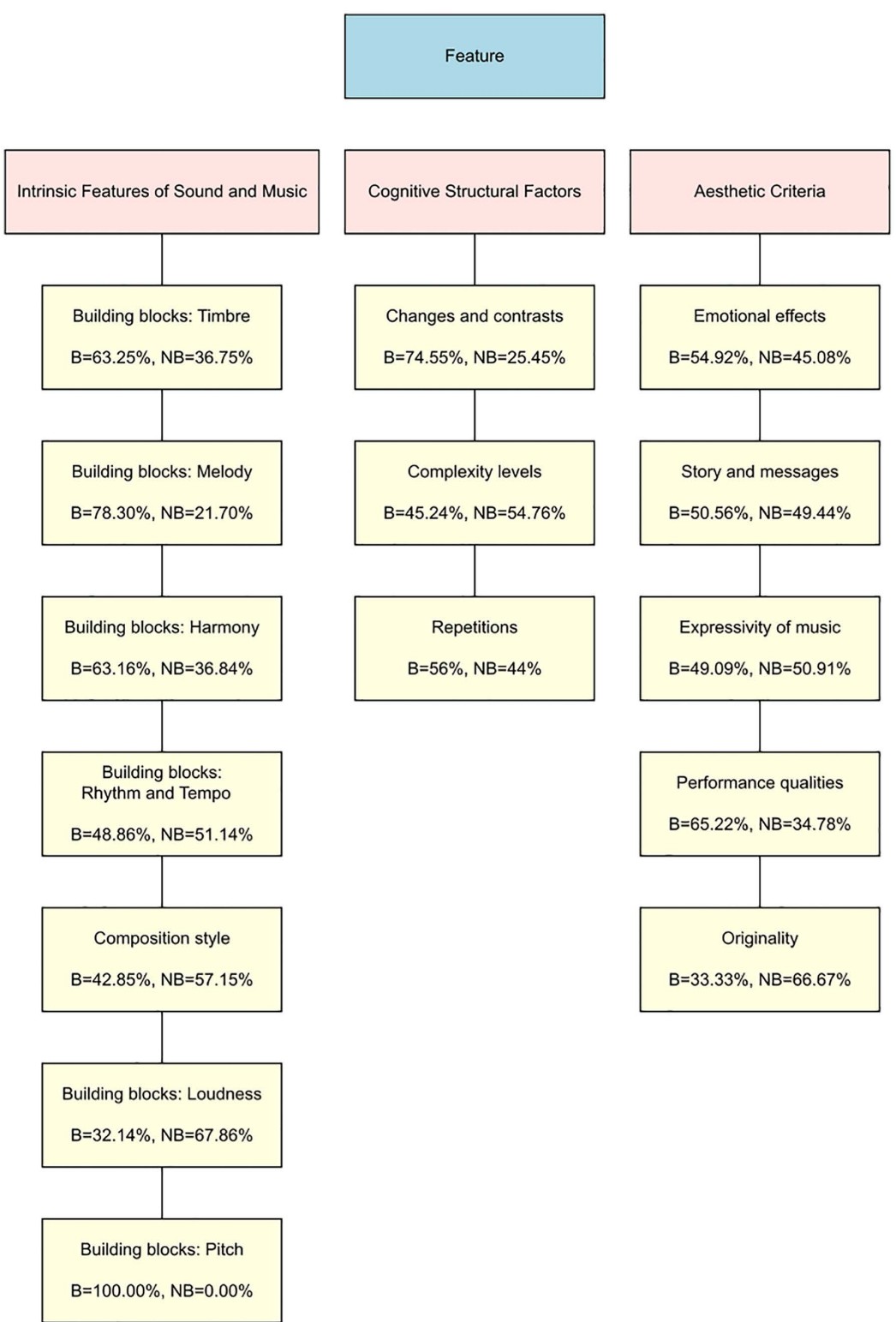

**Fig 2. Themes and codes derived from responses to Feature question.** The three boxes in pink show three themes from answers to the Feature question, while the yellow boxes below each of them show their constituent codes. In each code box, the percentage of times that code was assigned to beautiful (B) and not beautiful pieces (NB) responses is shown below each code name.

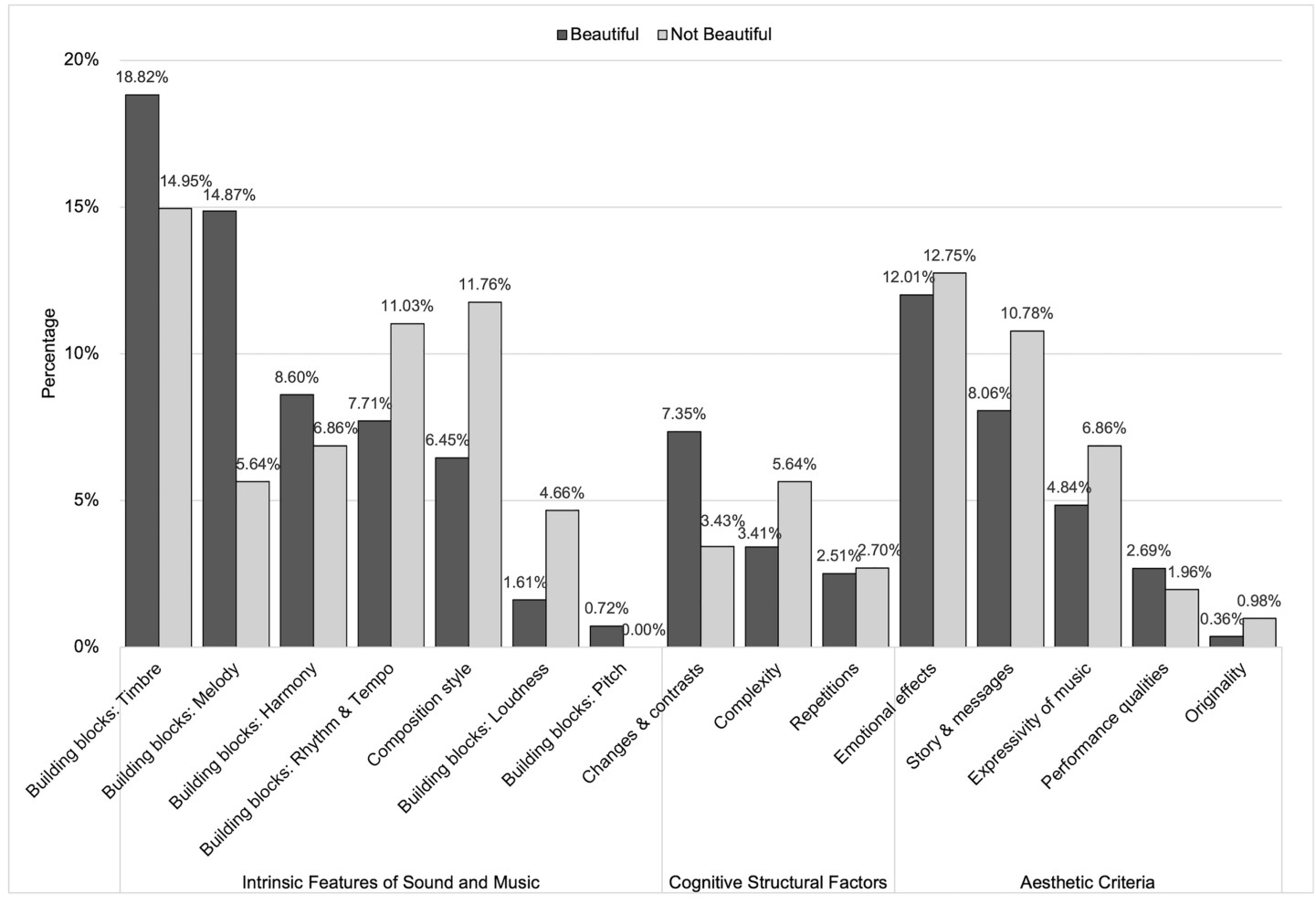

**Fig 3. Prominence of each code in responses describing contributing musical features.** The distribution of codes in response to the Feature question are shown in descending order based on the percentage of each code in the beautiful category. The dark and light grey bars represent the beautiful and not beautiful categories, respectively.

More related to the music, participants' responses also revealed that pieces that they considered beautiful tend to be expressive (F-"Expressivity of Music"-B16), while pieces without expressivity (F-"Expressivity of Music"-NB16a) or inducing too much arousal (F-"Expressivity of Music"-NB16b) could not be beautiful. A few participants explained that a style of performance enhanced or reduced its beauty (F-"Performance Qualities"-B17, F-"Performance Qualities"-NB17) and that originality—distinctive or unique features in sound and style—set beautiful pieces apart from others (F-"Originality"-B18, F-"Originality"-NB18).

### Findings from thematic analysis: Affect

Regarding the Affect questions, 16 codes were identified, from which six themes emerged (Figs 4 and 5). Fig 4 shows the relative percentages of responses for beautiful and not beautiful music for each code. Fig 5 shows, for beautiful and not beautiful music separately, the number of times a given code was assigned relative to the total number of times all codes were assigned for the Affect question. Examples of quotes for each code in each theme are presented in Table 2.

**Table 1. Examples of responses to Feature questions.**

| Theme | Quotes (Quote IDs) | |
|---|---|---|
| | **Beautiful music** | **Not beautiful music** |
| **Intrinsic Features of Sound and Music** | beautiful voice, rich tone (F-"Building Blocks: Timbre"-B1) | The vocal is quite raspy and rough (F-"Building Blocks: Timbre"-NB1) |
| | I find them emotive with melodies that are pleasing to the ear. (F-"Building Blocks: Melody"-B2) | melody is disjointed (F-"Building Blocks: Melody"-NB2) |
| | "pretty" sounding harmonies with some well-fitting dissonance (F-"Building Blocks: Harmony"-B3) | there's no harmony or melody to speak of (F-"Building Blocks: Harmony"-NB3) |
| | The slow tempo, the long-held notes, the instruments (F-"Building Blocks: Rhythm and Tempo"-B4) | The song is fast/sound is fast. Fast music sometimes increases pulse rate, although it sounds nice. (F-"Building Blocks: Rhythm and Tempo"-NB4) |
| | Soft-sounding, delicate vocals and piano (F-"Building Blocks: Loudness"-B5) | some of tone in the piece is too loud (F-"Building Blocks: Loudness"-NB5) |
| | It starts very subdued and builds beautifully when it gets to the line "Exsultemus, et in ipso jucundemur" as it moves from major to minor, the pitch goes up (F-"Building Blocks: Pitch"-B6) | N/A[a] |
| | The way the music is composed and all the instruments fit together perfectly (F- "Composition Style"-B7) | Sound like technical studies in the first movement (F-"Composition Style"-NB7) |
| | It's easier to find classical pieces that I consider beautiful compared to songs from other genres. (F-"Composition Style"-B8) | I don't think punk rock music is beautiful even when it's good. (F-"Composition Style"-NB8) |
| **Cognitive Structural Factors** | At verse 3, the guitar changes from finger picking to strumming, and a kick drum is introduced. The tempo picks up and new instruments are introduced. What makes it beautiful is the contrast from how gentle the song is in the beginning, to the large climax at the end. (F-"Changes and Contrasts"-B9) | Too simple and straightforward, lacks a bit of dramatic contrasts (F-"Changes and Contrasts"-NB9a) |
| | | Melodies tend to be fragmented, frequent changes between major and minor keys maintain the restlessness (F-"Changes and Contrasts"-NB9b) |
| | The production spaces out these lines so they don't sound muddy and you can hear each individual part. (F-"Complexity"-B10) | The arrangement is too simple. (F-"Complexity"-NB10a) |
| | | I think this music contains a great deal more notes, more instruments and less space for pause and breath. (F-"Complexity"-NB10b) |
| | The repeated phrases that run through each movement create a sense of connection and link the whole piece together so that it flows and creates the most beautiful images and stories in our mind. (F-"Repetitions"-B11) | I find the repetition and vocal effects a little annoying. (F-"Repetitions"-NB11) |
| **Aesthetic Criteria** | I can't hear this piece of music without being moved to tears. (F-"Emotional Effects" -B12) | they energise me and make me smile but don't necessarily touch my soul in a way that the pieces I described as beautiful do. (F-"Emotional Effects"-NB12) |
| | The content and lyrics principally, referring to extremely sad subjects, but still able to maintain hope. (F-"Story and Messages"-B13) | it is a song about how men mistreat women, I would not describe this as a beautiful song but more of an angry song. (F-"Story and Messages"-NB13) |
| | Lyrics are profound. Makes you relate to the artist. (F-"Story and Messages"-B14) | Aggressive, dark themes, not relaxing (F-"Story and Messages"-NB14) |
| | Story Telling was synchronous with the sound. (F-"Story and Messages"-B15) | Words don't flow very nicely with the composition of the music. (F-"Story and Messages"-NB15) |

*(Continued)*

**Table 1.** (Continued)

| Theme | Quotes (Quote IDs) | |
|---|---|---|
| | **Beautiful music** | **Not beautiful music** |
| | Joni Mitchell's voice and her capacity to capture deep emotion in her song writing. (F-"Expressivity of Music"-B16) | Melody not as expressive as it could be (F-"Expressivity of Music"-NB16a) |
| | | it has a lot of energy, the level of arousal is really high, and it is almost frightening. (F-"Expressivity of Music"-NB16b) |
| | The performances in this song are just so so passionate. You almost notice something new every time you listen to it. (F-"Performance Qualities"-B17) | Music that I wouldn't describe as beautiful mostly comes down to …, more of an emphasis on catchy or energetic performances. (F-"Performance Qualities"-NB17) |
| | I find that each song has their own, beautiful landscape. They use space, including stereo imaging, very creatively. The songs also make use of various techniques, such as sustained notes/chords and arpeggios to create unique and gripping atmospheres. (F-"Originality"-B18) | Generic, catchy, unoriginal (F-"Originality"-NB18) |

[a]There were no text responses to not beautiful pieces referencing Building block: Pitch.

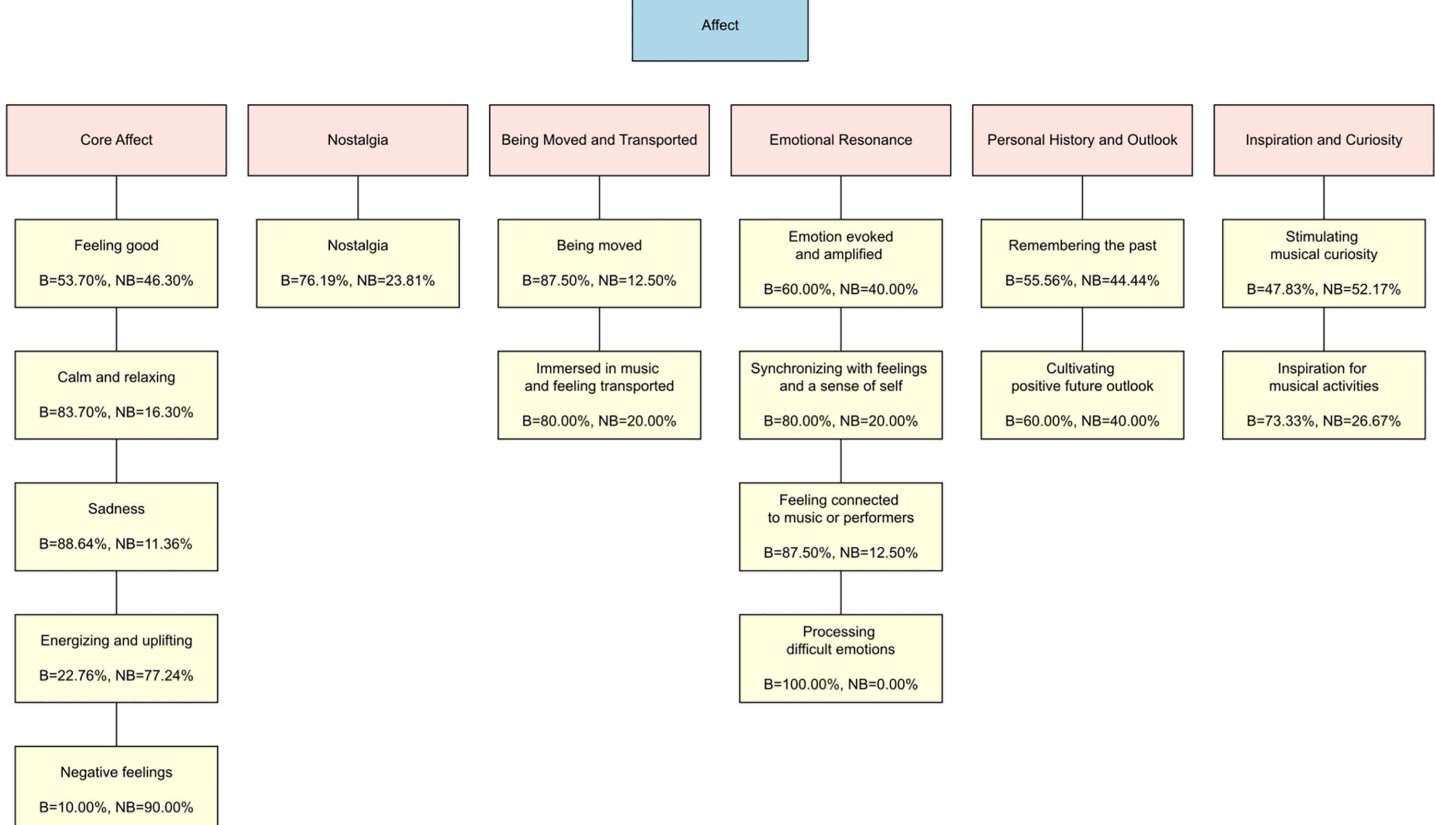

**Fig 4. Themes and codes derived from responses to Affect questions.** The three boxes in pink show three themes from responses to the Affect questions, while the yellow boxes below each of them show their constituent codes. In each code box, the percentage of times that code was assigned to beautiful (B) and not beautiful pieces (NB) responses is shown below each code name.

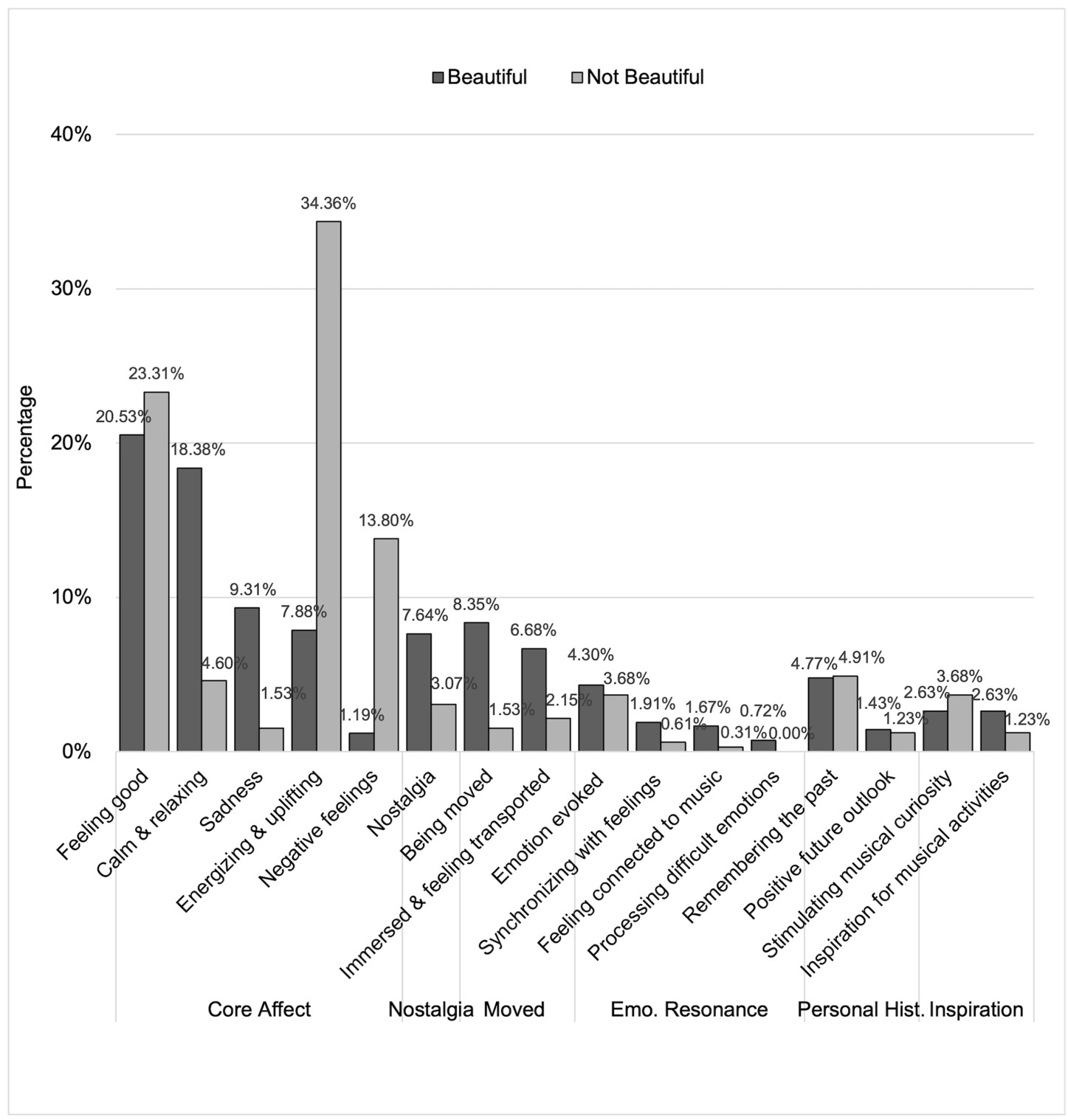

**Fig 5. Prominence of each code in responses describing experienced affect.** The distributions of codes in response to the Affect question are shown in descending order based on the percentage of each code in the beautiful category. The dark and light grey bars represent the beautiful and not beautiful categories, respectively.

Wherever relevant, '(Beautiful = x %, Not Beautiful = x %)' in the text denotes what percentage of assignations were in response to beautiful and not beautiful pieces respectively within each code.

**Core affect.** Responses revealed that general, positive feelings such as happy or good are often experienced from both types of pieces (A-"Feeling Good"-B1, A-"Feeling Good"-NB1). However, participants tended to experience positive valence and low arousal emotions (such as calm, relaxing, peaceful and contemplative) from beautiful pieces (A-"Calm and Relaxing"-B2, A-"Calm and Relaxing"-NB2, Beautiful = 83.70%, Not Beautiful = 16.30%) and, conversely, positive valence and high arousal emotions (such as energising, uplifting and upbeat) from not beautiful pieces (A-"Energizing and Uplifting"-B3, A-"Energizing and Uplifting"-NB3, Beautiful = 22.76%, Not Beautiful = 77.24%). Some mentioned experiencing sadness from both types of pieces (A-"Sadness"-B4, A-"Sadness"-NB4), but it was more commonly experienced from pieces participants reported as beautiful (Beautiful = 88.64%, Not Beautiful = 11.36%). In addition, participants' comments revealed that while the sadness experienced from pieces found beautiful tended to accompany mixed valence feelings such as "pleasurable sorrow", the sadness from pieces not considered beautiful were more intrinsically negative with words like "depressed" being used. Lastly, participants sometimes reported experiencing negative feelings such as a sense of restlessness and agitation when listening to pieces they do not find beautiful (A-"Negative Feelings"-NB5), but rarely when listening to pieces they find beautiful (10% in our sample, A-"Negative Feelings"-B5 and with negative emotions being related to angst and frustration).

**Nostalgia and being moved and transported.** Participants described experiencing nostalgia for both types of pieces (A-"Nostalgia"-B6, A-"Nostalgia"-NB6), but more commonly for pieces considered beautiful (Beautiful = 76.19%, Not Beautiful = 23.81%). Participants also reported that, when listening to music, they experienced deep emotional states such as being moved by music (A-"Being Moved"-B7, A-"Being Moved"-NB7), being immersed in the music and forgetting reality (A-"Immersed"-B8, A-"Immersed"-NB8), and feeling transported to a different place or time (A-"Immersed"-B9, A-"Immersed"-NB9). Such experiences were particularly evident in responses to pieces considered beautiful (87.50% for *being moved*, 80.00% for *immersed in music and feeling transported*), but were also experienced when listening to pieces perceived as not beautiful (12.50% for *being moved*, 20.00% for *immersed in music and feeling transported*). Many participants described physiological responses such as crying, goosebumps, shivers, and chills, as co-occurring with the experience of being moved or feeling transported (A-"Being Moved"-B10, A-"Being Moved"-NB10).

**Emotional resonance.** Responses from participants revealed that they have a strong personal and emotional connection with pieces of music they listen to often, and particularly those considered beautiful (the percentages of responses to beautiful pieces ranged from 60 to 100%). Participants reported both beautiful and not beautiful music as being able to evoke, stir and amplify emotions being experienced (A-"Emotion Evoked"-B11, A-"Emotion Evoked"-NB11, Beautiful = 60.00%, Not Beautiful = 40.00%), but it was more common that beautiful music would lead participants to report that they felt themselves or recognised their own emotions in what the music expresses (A-"Synchronizing with Feelings"-B12, A-"Synchronizing with Feelings"-NB12, Beautiful = 80.00%, Not Beautiful = 20.00%). Participants felt, perhaps as a result, connected to the music or to the performers (A-"Connected to Music"-B13, but see not beautiful pieces: A-"Connected to Music"-NB13, Beautiful = 87.50%, Not Beautiful = 12.50%). Finally, it was only for beautiful pieces that participants reported using the music to help them cope with difficult emotions such as the pain of loss or grief (A-"Processing Difficult Emotions"-B14), (Beautiful = 100.00%, Not Beautiful = 0.00%).

**Personal history and outlook.** Participants' responses demonstrated the extent to which music—perceived as both beautiful and not beautiful—is integrated with their lives, both past and future. Many reported that listening to both types of music reminded them of their lives, specific events or people in the past, and this to a largely equal extent in beautiful and not beautiful pieces (A-"Remembering the Past"-B15, A-"Remembering the Past"-NB15, Beautiful = 55.56%, Not Beautiful = 44.44%). Similarly, participants mentioned that listening to both beautiful and not beautiful tended to make them feel positive about their lives or world (A-"Positive Future Outlook"-B16, A-"Positive Future Outlook"-NB16, Beautiful = 60.00%, Not Beautiful = 40.00%).

**Table 2. Examples of responses to Affect questions.**

| Theme | Quotes (Quote IDs) | |
|---|---|---|
| | **Beautiful music** | **Not beautiful music** |
| **Core Affect** | They make me feel happy. (A-"Feeling Good"-B1) | I feel on top of the world listen to these music every time. (A-"Feeling Good"-NB1) |
| | the piano in this is beautiful and makes me feel like content. (A-"Calm and Relaxing"-B2) | This song makes me feel relaxed. (A-"Calm and Relaxing"-NB2) |
| | It makes me feel excited and full of energy every time I listen to it. (A-"Energizing and Uplifting"-B3) | The heavy guitars and raw vocals of this song feel like pumping adrenaline straight into your veins. (A-"Energizing and Uplifting"-NB3) |
| | Sad, i can just imagine looking out of a window on a rainy day and letting the sound overwhelm you. (A-"Sadness"-B4) | A dissonant chord as begin, sad, heavy (A-"Sadness"-NB4) |
| | I feel a sense of angst and dread, like an existential crisis. (A-"Negative Feelings"-B5) | I feel agitated and restless when I listen to this song. (A-"Negative Feelings-NB5) |
| **Nostalgia** | I feel nostalgic and hopeful at the same time. Every time I listen to this I am brought back to the first time I heard it. (A-"Nostalgia"-B6) | It evokes feelings of nostalgia. (A-"Nostalgia"-NB6) |
| **Being Moved and Transported** | stirs big feelings (A-"Being Moved"-B7) | Moved like I want to sing loud and hear the message (A-"Being Moved"-NB7) |
| | I tend to forget about everything surrounding me and I can only focus on these tracks, being totally absorbed by them. (A-"Immersed"-B8) | lost in the moment (A-"Immersed"-NB8) |
| | All of the piece is comfortable and allows me to go into imaginary places. (A-"Immersed"-B9) | I feel like rocking out and imagine being at a concert with these bands to enjoy it live and go nuts. (A-"Immersed"-NB9) |
| | I just feel the music deeply so that the hairs stand up or I get goosebumps. (A-"Being Moved"-B10) | These pieces of music sometimes give me shivers as they are really fantastic. (A-"Being Moved"-NB10) |
| **Emotional Resonance** | Because I regard them as beautiful pieces of music, if I listen to them when I am experiencing emotions of sadness, they will amplify the sadness immensely, and allow me to feel my feelings and in a way which is cathartic. (A-"Emotion Evoked"-B11) | Typical Mahler; excessively agonising and long drawn out, very emotional, dragging the listener through an emotional ringer. (A-"Emotion Evoked"-NB11) |
| | I feel a sense of oneness when listening to these songs. (A-"Synchronizing with Feelings"-B12) | it decries my own emotions. (A-"Synchronizing with Feelings"-NB12) |
| | The powerful vocals and emotional lyrics in "Heal the World" and "The Power of Love" resonate with me, creating a deep connection to the music and the artists. (A-"Connected to Music"-B13) | Two Fingers I'd listen to for reliability and a storyline I can relate to. (A-"Connected to Music"-NB13) |
| | It brings structure to my grief and allows me to express it. (A-"Processing Difficult Emotions"- B14) | N/A[a] |
| **Personal History and Outlook** | Thinking of people who are no longer with me fondly (A-"Remembering the Past"-B15) | This song reminds me of my teenage years, and there is a sense of reminiscence. (A-"Remembering the Past"-NB15) |
| | It just kind of makes me feel that everything is OK in the world and not to worry. (A-"Positive Future Outlook"-B16) | makes me believe love makes the world peaceful (A-"Positive Future Outlook"-NB16) |
| **Inspiration and Curiosity** | The fusion of jazz, funk, and rock elements creates a sonic tapestry that's both intellectually stimulating and emotionally gripping. (A-"Stimulating Musical Curiosity"-B17) | Energised, mentally active my instinct is to track the rhythms/ layering/ instrumentation etc (A-"Stimulating Musical Curiosity"-NB17) |
| | This piece … inspires me to grab my guitar and start playing along. (A-"Inspiration for Musical Activities"-B18) | Dazzled by the ability of the player, stimulated in the sense that I want to look for clues that I can use in my own practice & teaching of the piece (A-"Inspiration for Musical Activities"-NB18) |

[a]There was no responses that suggested not beautiful music helps the process of grieving.

**Inspiration and curiosity.** Finally, some participants, especially those who were musicians, commented on how listening both to beautiful and not beautiful pieces musically influenced them. Interestingly though, while both were found intellectually interesting and satisfying, and stimulated their musical intelligence and curiosity (A-"Stimulating Musical Curiosity"-B17, A-"Stimulating Musical Curiosity"-NB17, Beautiful = 47.83%, Not Beautiful = 52.17%), beautiful music tended to induce more reports of music motivating them to practise more, write more music, and explore more repertoire and performing styles (A-"Inspiration for Musical Activities"-B18, A-"Inspiration for Musical Activities"-NB18, Beautiful = 73.33%, Not Beautiful = 26.67%).

### Findings from thematic analysis: Impact

In response to the questions asking whether any of the listed pieces impacted the participants, the responses for beautiful pieces were: *Yes*, 76.54%; *No*, 16.05%; No answer, 7.41%. Among those who answered "No", 38.46% reported being impacted by other beautiful pieces not listed. For not beautiful pieces, the responses were: *Yes*, 61.73%; *No*, 30.86%; No answer, 7.41%. 25.00% of participants who answered "No" reported that they had been impacted by other not beautiful pieces.

Participants' descriptions of the impact of experiencing beauty led us to extract four themes from across 16 codes, which are shown in Fig 6. Fig 6 also shows the relative percentages of responses for beautiful and not beautiful music for each code. Fig 7 shows the number of times a given code was assigned relative to the total number of times all codes were assigned for the Impact question for beautiful and not beautiful music separately. Table 3 presents some of the responses that support each theme. Impacts ranged from more short-lived (e.g., *Affect while listening*) to more long-term (e.g., *Musical influence*).

**Affect while listening.** Many participants mentioned the experienced emotions as one of the impacts of finding a piece beautiful (I-"Feeling Good"-B1) or not beautiful (I-"Feeling Good"-NB1). The emotions reported were similar to those mentioned in the Affect questions, such as general positive feelings (I-"Feeling Good"-B1, I-"Feeling Good"-NB1, Beautiful = 62.50%, Not Beautiful = 37.50%), and calm and relaxation mainly from beautiful music (I-"Calm and Relaxing"-B2, I-"Calm and Relaxing"-NB2, Beautiful = 90.00%, Not Beautiful = 10.00%). Emotions being evoked and stirred in general was also reported, particularly from beautiful music (I-"Emotion Evoked"-B3, I-"Emotion Evoked"-NB3, Beautiful = 87.50%, Not Beautiful = 12.50%) as were feeling of being moved or immersed in music (I-"Being Moved"-B4, I-"Being Moved"-NB4, Beautiful = 75.00%, Not Beautiful = 25.00%). Only not beautiful pieces resulted in reports of negative feelings (I-"Negative Feelings"-NB5), energising and upbeat feelings (I-"Energizing and Uplifting"-NB6), and nostalgia (I-"Nostalgia"-NB7).

**Psychological impact.** Participant responses showed that listening to both beautiful and not beautiful pieces may have long-lasting emotional and psychological impacts, albeit to a greater extent for beautiful music. Beautiful music was revealed to act as a stronger emotional support for listeners in a vulnerable state (Beautiful = 77.28%, Not Beautiful = 22.72%, I-"Emotional Help"-B8, I-"Emotional Help"-NB8). Further, listening to beautiful music impacted the listener by changing their mood or emotions in a positive way (I-"Mood and Feelings"-B9a, I-"Mood and Feelings"-NB9, Beautiful = 65.22%, Not Beautiful = 34.78%), and by helping them understand themselves (I-"Mood and Feelings"-B9b). Interestingly, two responses out of 61 in this theme related how the impact of listening to pieces of music can extend beyond the self, offering opportunities to learn about different cultures and different perspectives on life (I-"Understanding Emotions"-B10, I-"Understanding Emotions"-NB10, Beautiful = 75%, Not Beautiful = 25%).

**Memory and the self.** Participants' comments demonstrated that while both beautiful and not beautiful music helped them connect with their lives, not beautiful music showed a slightly higher tendency for codes related to everyday use and identity (57.14% to 100%). Participants described how both beautiful and not beautiful pieces led them to relive memories (I-"Remembering the Past"-B11, I-"Remembering the Past"-NB11) of past situations

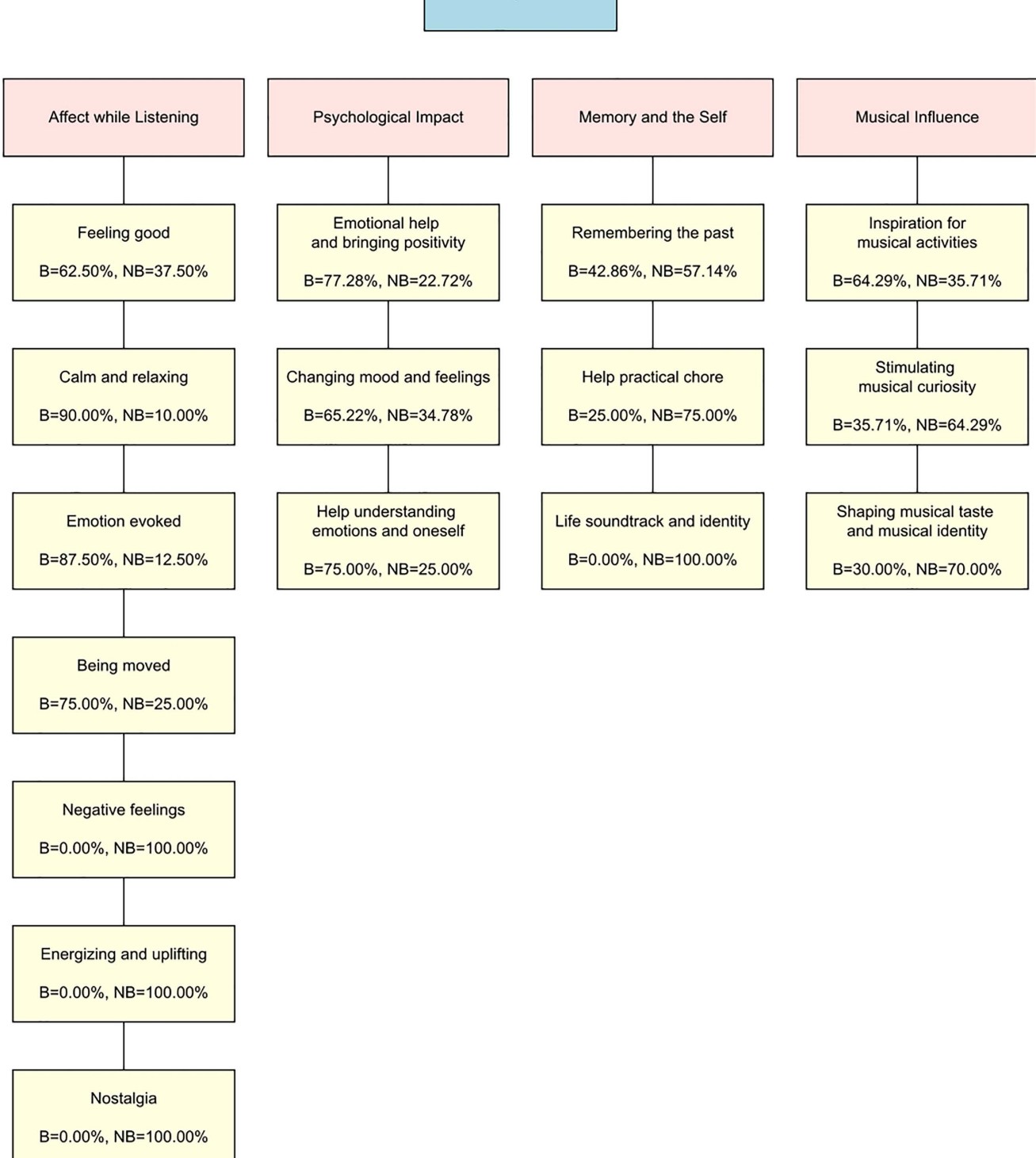

**Fig 6. Themes and codes derived from responses to Impact questions.** The three boxes in pink show three themes from responses to the Impact questions, while the yellow boxes below each of them show their constituent codes. In each code box, the percentage of times that code was assigned to beautiful (B) and not beautiful pieces (NB) responses is shown below each code name.

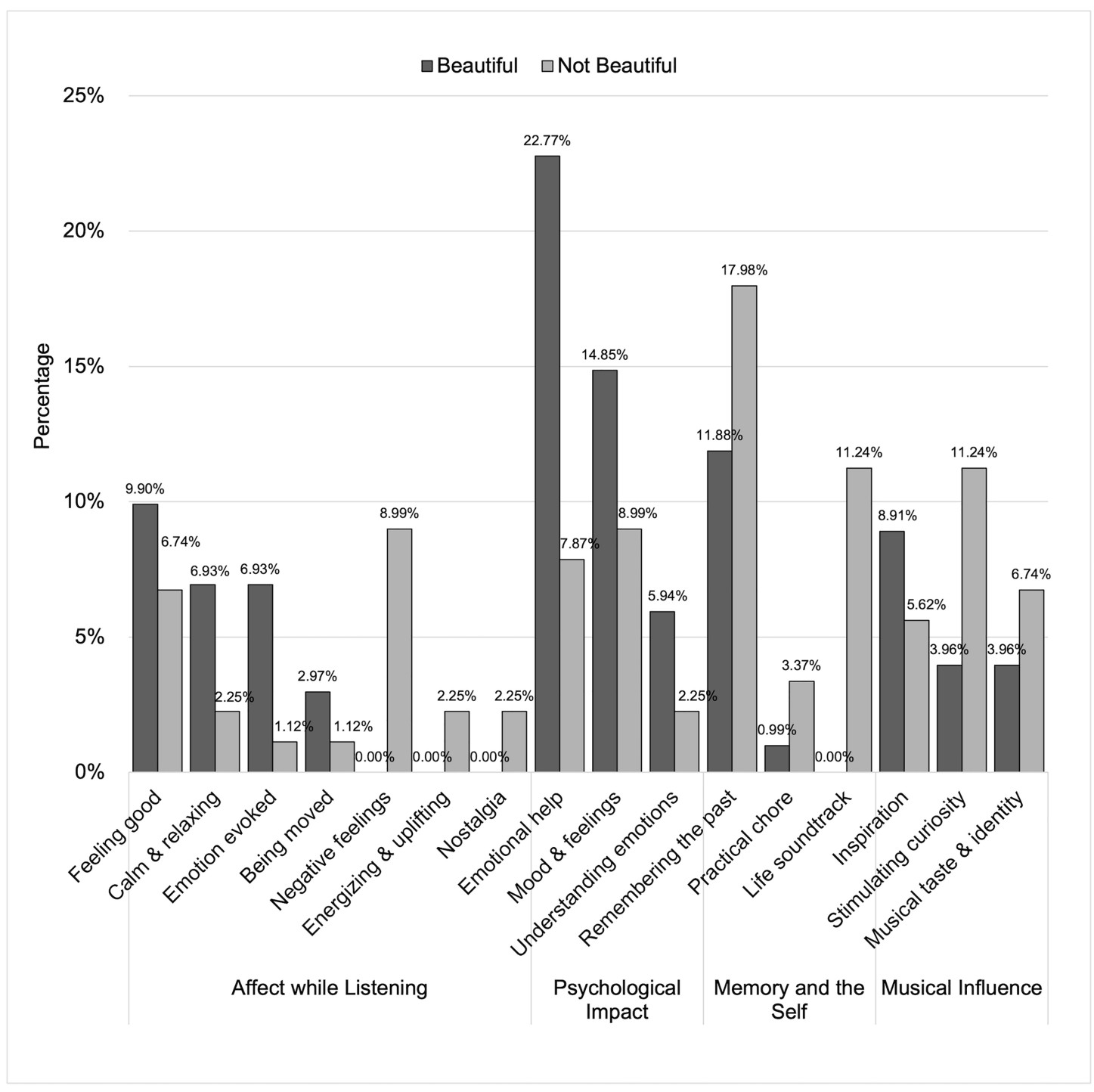

**Fig 7. Prominence of each code in responses describing impact from beautiful and not beautiful music.** The distribution of codes in response to the Impact question are shown in descending order based on the percentage of each code in the beautiful category. The dark and light grey bars represent the beautiful and not beautiful categories, respectively.

**Table 3. Examples of responses to Impact questions.**

| Theme | Quotes (Quotes ID) | |
|---|---|---|
| | **Beautiful music** | **Not beautiful music** |
| **Affect while Listening** | They all make me happy. (I-"Feeling Good"-B1) | They impact me in a positive way as both a musician and a listener as they make me feel good. (I-"Feeling Good"-NB1) |
| | They bring me peace, and reflection. (I-"Calm and Relaxing"-B2) | make me feel warm and at peace, even if they are sonically unremarkable. (I-"Calm and Relaxing"-NB2) |
| | they all inspire emotions and feelings inside me. (I-"Emotion Evoked"-B3) | They unlock different feelings. (I-"Emotion Evoked"-NB3) |
| | they really touch the soul in a way that can't be described. (I-"Being Moved"-B4) | I get lost in a song's lyrics. (I-"Being Moved"-NB4) |
| | N/A[a] | Certain pieces like Brides to Darkness creep me out but not in a bad way. (I-"Negative Feelings"-NB5) |
| | N/A[b] | Energised (I-"Energizing and Uplifting"-NB6) |
| | N/A[c] | The music itself creates nostalgia when listened to now. (I-"Nostalgia"-NB7) |
| **Psychological Impact** | These pieces of music are part of me. They have repaired and sustained my mental health. (I-"Emotional Help"-B8) | They help me get through tough times. (I-"Emotional Help"-NB8) |
| | Generally if I'm not in a good mood listening to any of these songs will help to change my mood. (I-"Mood and Feelings"-B9a) | They have all helped me at one time or another to change my mood in a positive sense. (I-"Mood and Feelings"-NB9) |
| | When listening to songs like the 3 mentioned, it often transports me to a much more emotional mindset – where I often reflect on my past, both the positives and negatives. This impacts me because I feel like I am able to feel emotions more deeply and review past actions of mine – therefore impacting my future. (I-"Mood and Feelings"-B9b) | |
| | Yes. Cultural orientation. Learning people's culture through their song. (I-"Understanding Emotions"-B10) | Bowie's songs always push and challenge me to look at the world differently. (I-"Understanding Emotions"-NB10) |
| **Memory and the Self** | They remind of good times when I have listened to them previously. (I-"Remembering the Past"-B11) | it reminds me of friendships, as I got to know Tom Waits' music through a friend's parents. (I-"Remembering the Past"-NB11) |
| | Music that I like very much has also connected itself to troubled times and whilst I want to still listen I find I struggle to do so as the memories are provoked rather too much. I never want to hear Ruby Ruby Ruby again. (I-"Remembering the Past"-B12) | Swan Lake brought back bad memories as its certain passages conveys tragedy elements. (I-"Remembering the Past"-NB12) |
| | I do listen to other pieces of music when trying to sleep – so the alpha waves, theta waves and Tibetan singing bowls impacts me by having a profound ability to help with my sleep. (I-"Practical Chore"-B13) | Gets my adrenaline going and increases my engagement with anything mundane that I might be doing around the house (I-"Practical Chore"-NB13) |
| | N/A[d] | Although I'm not as interested in them [the Beatles and Rolling Stones] as I used to be, I still consider them to have had a great impact in shaping my personality and my music taste. (I-"Life Soundtrack"-NB14) |

*(Continued)*

**Table 3.** (Continued)

| Theme | Quotes (Quotes ID) | |
|---|---|---|
| | **Beautiful music** | **Not beautiful music** |
| **Musical Influence** | The main impact is to want to listen to them regularly and to hear more details in them at different times. (I-"Inspiration"-B15a) | Television's music has made me re-assess how I play the guitar. (I-"Inspiration"-NB15) |
| | listening to beautiful piano music like this makes me want to improve on my own piano skills. (I-"Inspiration"-B15b) | |
| | These pieces have also impacted my compositional style, as I really wanted to understand how to capture this feeling in sound the way these composers did. (I-"Inspiration"-B16) | They.. show me different ways to express yourself through music, with different emotions such as anger and angst that can be communicated in a song. (I-"Inspiration"-NB16) |
| | Inspired me to follow music as a career (I-"Inspiration"-B17) | inspiration to pursue music as a large part of my life (I-"Inspiration"-NB17) |
| | Of the music around me in my childhood home it was the first movement of 'Spring' from the Vivaldi 'Four Seasons' that attracted my attention and introduced me to a love of classical music. (I-"Stimulating Curiosity"-B18) | As a violin student, the Schoenberg was a revelation to me. It extended my notion of what the instrument was capable of and what truly virtuosic playing was really about! (I-"Stimulating Curiosity"-NB18) |
| | I grew up with them and they definitely shaped my music taste forever. (I-"Musical Taste and Identity"-B19) | All three songs have changed my life in one way or another, they are all key songs in my ever-developing music taste. (I-"Musical Taste and Identity"-NB19) |
| | They also generally made me the musical person I am nowadays. (I-"Musical Taste and Identity"-B20) | These artists are part of my identity as a musician. (I-"Musical Taste and Identity"-NB20) |

a, b, c No responses to beautiful pieces referred to feeling energised (a), feeling nostalgia (b), or experiencing negative emotions (c).

d We did not find any response mentioning beautiful music's role in identity formation.

in which they listened to and/or performed the piece (I-"Remembering the Past"-B11), and were associated with memories that were not always positive (I-"Remembering the Past"-B12, I-"Remembering the Past"-NB12). However, not beautiful music especially was described as serving as a practical aid to daily life activities, from housework, to the daily commute, and even to sleep (I-"Practical Chore"-B13, I-"Practical Chore"-NB13). Similarly, not beautiful music exclusively was described as having shaped the participants' identity or personality (I-"Life Soundtrack"-NB14, Beautiful = 0%, Not Beautiful = 100%).

**Musical influence.** Beautiful pieces were more frequently mentioned as having motivated the listener to engage further with music (Beautiful = 64.29%, Not Beautiful = 35.71%), such as by listening to the piece more frequently (I-"Inspiration"-B15a), and working to improve their performance skills (I-"Inspiration"-B15b, I-"Inspiration"-NB15). However, both beautiful and not beautiful music were reported to have given them a deeper understanding of how to be more expressive (I-"Inspiration"-B16, I-"Inspiration"-NB16), and to have encouraged them to become a professional musician (I-"Inspiration"-B17, I-"Inspiration"-NB17).

Finally, it is important to note that participants reported not beautiful music especially (Beautiful = 35.71%, Not Beautiful = 64.29%) as having stimulated their musical curiosity (I-"Stimulating Curiosity"-NB18) and as having widened their repertoire or interest (I-"Stimulating Curiosity"-B18). It is also notable that (similar to beautiful music never being reported as having shaped personal identity (see *Memory and the self*)), there was a greater tendency for not beautiful, compared to beautiful (Beautiful = 30.00%, Not Beautiful = 70.00%), to be reported as having shaped the participants' musical taste (I-"Musical Taste and Identity"-B19, I-"Musical Taste and Identity"-NB19) and identity (I-"Musical Taste and Identity"-B20, I-"Musical Taste and Identity"-NB20).

## Quantitative analysis examining role of individual differences

Poisson mixed effects models were conducted to examine the influence of musicianship and personality traits on participants' descriptions of their experience of beauty, using the number of distinct codes identified within each theme within each question as dependent variables. For each question, a first, simpler model included musicianship (low, medium, high) and scores on the five personality traits as fixed effects, with participant as a random effect, while a second, more complex model included theme as an additional fixed effect.

**Feature.** Table 4 shows the results of the simple model for the Feature question, which revealed two significant predictors (musicianship medium ($B=0.32$, $SE=0.13$, $z=2.39$, $p=0.02$) and openness to experience ($B=0.12$, $SE=0.05$, $z=2.21$, $p=0.03$)), indicating that amateur musicians and participants with higher openness to experience scores mentioned more varied aspects of musical beauty. A second Poisson mixed effects model that allowed theme as an additional fixed effect to interact with musicianship and scores on the five personality traits, showed no significant interaction between musicianship medium and each theme ($p>0.05$) or between openness to experience and each theme ($p>0.05$), demonstrating that the effects of musicianship medium and openness to experience were not dependent on different themes.

**Affect.** Poisson models were conducted to examine the association between the range of aspects participants reported in response to the Affect question and each predictor (musicianship level, personality traits, and theme for the second more complex model). As summarised in Table 5, the results from a first, simpler model showed no significant effects of any predictors ($p>0.05$), suggesting that individual differences did not influence the breadth with which participants described emotions and feelings experienced from musical beauty. Importantly, a model that allowed theme to interact with musicianship level and personality traits also showed no significant interactions ($p>0.05$), confirming that the null finding in the simple model is not due to different patterns of behaviour across themes.

**Impact.** Finally, Poisson mixed effects models were once again used to examine if and how musical training and personality traits influence the range of impacts that participants reported from the experience of musical beauty. Here, while a first model showed no significant associations between any predictors and the number of codes identified, the more complex model that allowed theme to interact with the different predictors pointed to an interaction between musicianship and theme ($p=0.03$), and between conscientiousness and theme ($p=0.04$) that warranted further exploration.

To examine the former interaction in detail, follow-up Poisson mixed effects models were conducted separately for each theme, with the results summarised in Table 6. Both musicianship medium and high were found to be significant predictors for the theme *Musical influence* (musicianship medium ($B=2.49$, $SE=1.11$, $z=2.23$, $p=0.03$) and high ($B=2.69$, $SE=1.12$, $z=2.41$, $p=0.02$)), indicating that participants with more musical training mentioned a wider range of musical influences

**Table 4. Results of a Poisson mixed effects model that examined the associations between each predictor (musicianship and five dimensions of TIPI) and the number of codes mentioned within the Feature question.**

| Predictors | Parameter estimates | | | | Confidential intervals | |
|---|---|---|---|---|---|---|
| | Estimate | SE | z | p | LL | UL |
| Intercept | −0.09 | 0.42 | −0.21 | 0.84 | −0.91 | 0.73 |
| **Musicianship: Medium** | **0.32** | **0.13** | **2.39** | **0.02** | **0.06** | **0.59** |
| Musicianship: High | 0.22 | 0.14 | 1.62 | 0.10 | −0.05 | 0.49 |
| **Openness** | **0.12** | **0.05** | **2.21** | **0.03** | **0.01** | **0.22** |
| Extraversion | −0.04 | 0.04 | −1.05 | 0.29 | −0.11 | 0.03 |
| Conscientiousness | −0.01 | 0.05 | −0.11 | 0.91 | −0.10 | 0.09 |
| Agreeableness | −0.08 | 0.05 | −1.56 | 0.12 | −0.18 | 0.02 |
| Emotional Stability | 0.06 | 0.04 | 1.48 | 0.14 | −0.02 | 0.14 |

**Table 5. Results of a Poisson mixed effects model that examined the associations between each predictor (musicianship and five dimensions of TIPI) and the number of codes mentioned within the Affect question.**

| Predictors | Parameter estimates | | | | Confidential intervals | |
|---|---|---|---|---|---|---|
| | Estimate | SE | z | p | LL | UL |
| Intercept | −0.56 | 0.46 | −1.21 | 0.22 | −1.47 | 0.34 |
| Musicianship: Medium | −0.13 | 0.15 | −0.88 | 0.38 | −0.43 | 0.17 |
| Musicianship: High | 0.09 | 0.15 | 0.59 | 0.55 | −0.20 | 0.38 |
| Openness | 0.10 | 0.06 | 1.68 | 0.09 | −0.02 | 0.22 |
| Extraversion | 0.03 | 0.04 | 0.74 | 0.46 | −0.05 | 0.11 |
| Conscientiousness | −0.06 | 0.05 | −1.05 | 0.29 | −0.16 | 0.05 |
| Agreeableness | −0.06 | 0.06 | −1.09 | 0.27 | −0.17 | 0.05 |
| Emotional Stability | 0.01 | 0.04 | 0.13 | 0.89 | −0.08 | 0.09 |

**Table 6. Results of follow-up Poisson mixed effects models examining the interaction between musicianship and theme, conducted separately for each theme within the Impact question.**

| Predictors | Affect while Listening | | Psychological Impact | | Memory and the Self | | Musical Influence | |
|---|---|---|---|---|---|---|---|---|
| | Estimate | p | Estimate | p | Estimate | p | Estimate | p |
| Intercept | −0.83 | 0.60 | −1.44 | 0.27 | −2.13 | 0.36 | −8.60 | 0.00 |
| **Musicianship: Medium** | 0.41 | 0.40 | −0.43 | 0.29 | 0.43 | 0.54 | **2.49** | **0.03** |
| **Musicianship: High** | −0.20 | 0.71 | −0.44 | 0.29 | −0.27 | 0.73 | **2.69** | **0.02** |
| Openness | −0.28 | 0.15 | 0.23 | 0.18 | 0.00 | 0.99 | −0.29 | 0.21 |
| Extraversion | −0.11 | 0.45 | 0.05 | 0.64 | −0.46 | 0.06 | 0.33 | 0.13 |
| **Conscientiousness** | −0.11 | 0.54 | −0.01 | 0.97 | −0.03 | 0.90 | **0.96** | **0.02** |
| Agreeableness | 0.21 | 0.34 | 0.07 | 0.63 | 0.19 | 0.54 | 0.00 | 0.99 |
| Emotional Stability | 0.22 | 0.17 | −0.22 | 0.07 | 0.18 | 0.45 | −0.08 | 0.71 |

than non-musicians. In contrast, neither a medium nor high level of musicianship was a significant predictor for the number of codes mentioned in the other three themes—*Affect while listening*, *Psychological impact*, and *Memory and the self*.

Regarding the interaction between conscientiousness and theme, a Poisson mixed effects model with the full interaction demonstrated that conscientiousness was positively associated with *Musical influence* (*B*=0.68, *SE*=0.29, *z*=2.32, *p*=0.02), but negatively associated with *Affect while listening* (*B*=−0.68, *SE*=0.34, *z*=−2.02, *p*=0.03), and *Psychological impact* (*B*=−0.76, *SE*=0.32, *z*=−2.38, *p*=0.02) (see Table 7).

In sum, results showed that amateur musicians provided richer detail about what makes music beautiful and that musicians more generally reported more ways in which beautiful music had influenced their musical activities. With regard to personality, more open participants also provided richer detail about what makes music beautiful (like amateur musicians), while conscientious individuals described beautiful music as having more influence on their musical activities (like musicians) but less affective and psychological impact on them. Poisson mixed effects models computed using the responses to not beautiful pieces for each question failed to show the above documented effects.

## Discussion

While the experience of musical beauty holds a prominent place in musical creativity, performance and listening [5–7], how it is experienced has been insufficiently explored in terms of the contributing musical features (*Feature*), the range and patterning of feelings listeners report (*Affect*), and the impact it has on the listener (*Impact*). Also relatively unclear was how all three vary depending on the listener's musical training and personality traits.

**Table 7. Results of a follow-up Poisson mixed effects model examining the interaction between conscientiousness and theme within the Impact question.**

| Predictors | Parameter estimates | | | | Confidential intervals | |
|---|---|---|---|---|---|---|
| | Estimate | SE | z | p | LL | UL |
| Intercept | −5.54 | 1.81 | −3.07 | 0.00 | −8.04 | −2.46 |
| **Conscientiousness** | **0.68** | **0.29** | **2.32** | **0.02** | **0.16** | **1.32** |
| **Affect while Listening** | **4.36** | **2.01** | **2.17** | **0.03** | **0.71** | **8.70** |
| **Psychological Impact** | **5.24** | **1.93** | **2.72** | **0.01** | **1.91** | **9.46** |
| Memory and the Self | 3.26 | 2.26 | 1.45 | 0.15 | −1.08 | 7.95 |
| **Conscientiousness \*Affect while Listening** | **−0.68** | **0.34** | **−2.02** | **0.03** | **−1.33** | **−0.07** |
| **Conscientiousness \*Psychological Impact** | **−0.76** | **0.32** | **−2.38** | **0.02** | **−1.44** | **−0.18** |
| Conscientiousness \*Memory and the Self | −0.61 | 0.38 | −1.61 | 0.11 | −1.39 | 0.09 |

Fig 8 summarises the findings across our research questions. With regard to musical features, our study extended previous work to reveal that while musical beauty is attributable to certain *Intrinsic features of the sound and music*, it is also related to *Cognitive structural factors* such as the degree of complexity, change and repetition, and to *Aesthetic criteria* the listeners apply including *Expressivity of the music* and *Performance qualities*. Further, with regard to emotion and impact, our study showed that musical beauty evokes *Calm and relaxing* feelings, *Sadness*, and the feeling of *Being moved and transported*, in addition to having high *Emotional resonance* with the listener and the ability to inspire. Finally, it was shown that musical beauty uniquely impacts listeners primarily by providing *Emotional help and bringing positivity.* An exploratory quantitative approach showed that musicianship and openness to experience are positively associated with mentioning a greater array of contributing musical features, while musicianship and conscientiousness are related to more frequent mentions of musical beauty's impact on *Musical influence*.

These findings significantly expand our understanding of the experience of musical beauty by highlighting previously unexplored aspects of musical beauty, such as structural cognitive factors, epistemic emotions, long-term impact, and individual differences. In the following sections, we will discuss how our findings highlight the distinctive characteristics and occurrence of the experience of musical beauty.

### The intrinsic, structural, and aesthetic determinants of beautiful music

Many participants attributed their perception of beauty to fundamental features of music—a mellow timbre, a slow tempo, or a quiet sound—captured in the *Building blocks* codes. This is in line with theories that the aesthetic experience is at least partly based on the perception of the sensory features of stimuli [21,26]. Further, our results showed experiencing beauty from music is also informed by how the piece is composed (the code *Composition style*), such as the way in which a melody or chord progression develops, or the way in which voice and instrument parts blend together, as similarly reported in Fleckenstein et al. [12].

However, in addition to beautiful pieces having certain sonic features, our results showed that they are also characterised in terms of the degree of change, complexity, and repetition, as encompassed by the theme we called *Cognitive structural factors.* It is worth noting that these were all emphasised as having to be subjectively the right amount for the listener to experience beauty in music.

The role these factors play in the perception of musical beauty align with previous findings [13] showing that the experience of musical beauty may be associated with change in musical elements. It is also in line with theories on the mechanisms of aesthetic experience. For example, the contribution of changes and contrasts is consistent with the contrast effect: the idea that aesthetic experience occurs when stimuli with contrasting elements or contrasting affective characteristics appear successively [118–120], while the finding that the subjectively right level of complexity is required for beauty

**Fig 8. Summary of study findings.**

to be experienced aligns fully with the inverted-U theory [102,121] and, to some extent, with processing fluency theories [38]. Lastly, the idea that repeating patterns of melody, rhythm or chords give rise to beauty is in line with both the mere exposure effect [99] and the purported effect of the repetition of musical patterns on listener enjoyment [98,122]. To our knowledge, ours is the first study to show that these well-established cognitive mechanisms underlie listeners' evaluations of musical beauty.

Finally, a theme we labelled *Aesthetic criteria* because of its similarity to the criteria for aesthetic judgements listed by Juslin [3] encompassed *Emotional effects, Story and message, Expressivity of music, Performance quality,* and *Originality*. These reflected that a key determinant for listeners describing a musical piece as beautiful was whether the music led them to feel moved or connected with the music, as captured in the code *Emotional effects*, and as was reported in a study by Fleckenstein et al. [12]. Many participants described the emotional effects of beautiful music, even when they were being asked about the music's sonic features. This finding would seem to demonstrate the crucial role of the listener's emotions on their experience of beauty in music, corroborating the idea that experiencing beauty is largely an emotional experience [15,47,48].

Further, emphasis on the *Story and message* is in line with findings from previous studies on the important role of lyrics in strong experiences with music [8] (especially on evoking sad or melancholy emotions [123,124]), while emphases on the *Expressivity of music* (such as a "passionate" or "powerful" performance) and *Performance qualities* align with Fleckenstein et al. [12], which reports that listeners consider musical expressivity and performance skill or significance as contributing factors to perceived beauty. Finally, a few participants mentioned novelty or originality in pieces or performances as an influencing factor (the code *Originality*), thus at least partially supporting the idea in Juslin et al. [9] that "originality" is one of the most important criteria for aesthetic judgements. However, this code's low frequency highlights its relatively lesser importance compared to other features like *Emotional effects* and *Story and message*.

## The characteristic emotional profile of beautiful music

Our study demonstrated that beautiful and not beautiful pieces have distinctive emotional profiles, as encapsulated in the different themes. With regard to what we called, *Core affect*, listeners tended to experience low arousal emotions such as calm, relaxation, and feelings of peace more often when listening to beautiful pieces (the code *Calm and relaxing*), in line with a study on musical beauty and feelings [125] and another one on beauty in everyday life experiences [14]. In contrast, listeners experienced high arousal emotions such as energised or uplifted more often when listening to the not beautiful pieces (the code *Energizing and uplifting*). Negative emotions (such as restless, agitation, angst, hauntedness, frustration) were also reported, primarily in response to not beautiful pieces. The fact that participants chose to listen to music that evoked such emotions suggests that these pieces may hold aesthetic value despite not being considered beautiful and that experiencing these negative emotions through music may function as an emotional outlet [126,127].

Our participants also reported feelings of sadness in response to their beautiful pieces much more frequently than in response to the pieces they did not find beautiful (the code *Sadness*), in line with the positive associations between beauty and sadness reported in a questionnaire study [12] and in other empirical studies [52,54]. Findings from Peltola and Eerola [128] show that listeners experience different types of sadness from music, from a positive, sweet sorrow to a negative grief or melancholia. It is noteworthy that the description of the sadness experienced from beautiful pieces here had a more positive tone, compared to the negative tone of the sadness from not beautiful pieces.

The responses under the theme *Being moved and transported* also revealed that participants more frequently experience being moved by beautiful pieces. This is in line with the finding from Vuoskoski et al. [33] showing a positive correlation between musical beauty and the feeling of being moved. It is interesting to note here that the feeling of being moved has previously been argued to be a condition for the experience of beauty, as opposed to being the result of it. Indeed, Levinson [49] argues that the feeling of being moved is a critical determinant of musical beauty, and empirical evidence suggests that for beauty to be perceived from sad music, the feeling of being moved is critical [54]. In line with this, many participants from our questionnaire asserted that they tend to find a piece of music beautiful only if it moves them or "touches their soul".

In addition to the feeling of being moved, physical reactions (such as chills and goosebumps) and the feeling of being transported have previously been associated with musical beauty [12]. Our participant descriptions suggested that these experiences, along with the experience of awe—each considered a prototypical response to aesthetic experiences [23,129]— tend to occur more frequently when musical beauty is experienced. Moreover, participants reported that they experience the feeling of being connected to music and artists (i.e., a composer and/or performers) more frequently for beautiful music than for not beautiful music (the code *Feeling connected to music or performers* in the theme *Emotional resonance*). The feeling of being connected tends to co-occur with the feeling of being moved [33], and with the experience of chills from music [72]. Thus, that these aesthetic responses are more likely to happen when beauty is perceived in music suggests that musical beauty makes the experience of listening to music aesthetically and emotionally deeper and more intense.

However, perhaps the most striking aspect of the affective profile participants associated with beautiful music, and one which has not been reported elsewhere, is the emotional support it purportedly offers when people are experiencing emotional distress such as grief (the code *Processing difficult emotions* in the theme *Emotional resonance*). It has previously been reported that people use music to deal with emotional difficulties [63,130,131], to effectively regulate their mood [62,130], and to reconnect with their past or with deceased loved ones [63,130,131]. Our study demonstrates that people may tend to gain these kinds of emotional support particularly from pieces they find beautiful, echoing Levinson's suggestion [49] that beautiful music "provides some compensation for" (p. 134) emotional difficulties one may face in daily life.

There are arguably two reasons why the pieces that listeners find beautiful might help them to deal with emotional difficulties. The first pertains to the association between musical beauty and perceived sadness. Listening to sad music when grieving may enhance the mood of the bereaved [63], since if the listener can identify their emotion in the piece,

the emotional experience from sad music can become cathartic [63]. A second possible explanation is that when listening to pieces perceived as beautiful, the listener tends to experience feelings of being moved. Experiencing being moved releases psychological tension, especially when accompanied by the physiological reaction of tears [132]. Future studies may elucidate the functions of beautiful music by providing empirical evidence to confirm either or both of these explanations. Future studies could also explore the directionality of the associations: whether the music is consciously considered beautiful because it helps, or the music helps because the listener consciously finds it beautiful.

### The scope and the nature of beautiful music's impact

In line with the well-documented finding regarding the usage of music as an effective tool to manage and regulate one's own mood and emotions [62,133–135], many participants reported what could be considered shorter-term impacts, noting that listening to both beautiful and not beautiful pieces tended to change their affective state for the better, converting their negative mood or thoughts into positive ones (the code *Changing mood and feelings* in the theme *Psychological impact*). This was, however, more frequently reported for beautiful pieces, which may indicate that experiencing beauty creates conditions for the listener's mood to be changed more easily. Here, it is interesting to consider potential differences in how beautiful and not beautiful pieces change the listener's mood for the better. For beautiful pieces, this could be due to their soft and quiet sound, slow tempo and the resultant calmness, relaxation and consolation they bring, while for not beautiful pieces, this could be due to their louder sound, faster tempo and energising features, which may provide an emotional outlet for the listener.

There was a slight tendency for people to report that beautiful music, in contrast to not beautiful music, encouraged engagement in further musical activities, pointing to longer term effects of experiencing beauty in music. However, few other impacts from music were specific to beautiful music. For example, many mentioned that listening to both beautiful and not beautiful pieces enabled them to retrieve past events, people or feelings in memories, as captured in the code *Remembering the past*: an interesting addition to the wide literature on music and autobiographical memories, e.g., [136]. Similarly, listening to both beautiful and not beautiful pieces stimulated the listener's musical curiosity and intelligence—the desire to understand the structure of the piece, enjoy its complexity, discover novel musical expressions, or to engage in further musical activities. This study extends earlier work linking a desire to understand and a desire to continue the experience with beauty experiences [15,47], by showing that such epistemic emotions can also be evoked by not beautiful pieces.

Finally, it is interesting to note that some impacts seemed to be greater for not beautiful than for beautiful pieces. Specifically, participants reported that listening to not beautiful pieces influenced their musical preference for particular genres and helped shape their musical identity and identity in a broader sense (the code *Shaping musical taste and musical identity* in the theme *Musical influence*). Several studies have demonstrated music's contribution to identity formation [137–139] and so a lack of forming either a musical or broader identity around music one finds beautiful is interesting, in that it resonates with the philosophical idea of the universality of beauty, e.g., [44]. This idea, supported by previous empirical data [15], states that people experiencing beauty tend to feel that others will also find the piece they consider beautiful to be beautiful. The potential difficulty in distinguishing oneself from others through beautiful music may explain why the affordance of a longer-term identity is instead obtained from the other music people value and often choose to listen to.

### Individual differences in experiences of musical beauty

Finally, our exploratory, quantitative approach revealed that both amateur musicians (musicianship medium) and those who were high in openness to experience mentioned a more extensive range of features contributing to musical beauty, compared to participants with lower or higher musical expertise, and lower in openness to experience. This finding may be due to several factors: musicians' enhanced perceptual and cognitive musical abilities [82–87] and their greater ability

to explain music and musical experiences using technical terms [140]; and openness's positive association with musical cognitive skills [96], along with a greater tendency to articulate a greater variety of perceptual processes [141] and to use more complex language [142].

In any case, it is noteworthy that it was amateur rather than professional musicians (musicianship high) who mentioned a greater variety of features as contributing to musical beauty. Interestingly, amateur musicians also mentioned several codes in the theme *Aesthetic criteria* more frequently than professional musicians, as reflected in the percentage of relevant text responses out of the total number of responses that were seen at each musicianship level: *Story and messages* (amateur = 13.70%, professional = 8.52%), and *Performance qualities* (amateur = 6.37%, professional = 3.41%). The fact that professional musicians reported some of these codes less frequently may relate to the findings that individuals with more musical training tend to dissociate their felt emotions from the perception of music [88–90].

Another influence of musical expertise we observed is that, compared to non-musicians, both amateur and professional musicians reported a broader musical impact from the experience of musical beauty. This supports the idea that aesthetic musical experiences motivate some individuals to make a serious life commitment to music [64–66], and highlights a more domain-specific influence of aesthetic musical experiences. The finding extends the perspectives of prior research on the long-term impact of music [56] or aesthetic experience [27,55,57] that has predominantly focused on their influence on the emotional well-being, outlook, or social behaviour of the individual.

We observed that people high in conscientiousness mentioned the codes within the theme *Musical influence* more frequently, but the codes within the themes *Affect while listening* and *Psychological impact* less frequently compared to those low in this trait. It has been reported that conscientiousness is associated with trait inspiration and being curious [143–146] and our findings are thus interesting in suggesting that these tendencies can be observed in the long-term impact that the experience of musical beauty has on conscientious individuals.

Interestingly, no individual differences were found in terms of the number of codes identified in the Affect questions. This suggests that the range of emotions listeners experience in response to musical beauty are not as influenced by musical expertise or personality traits, which partially supports Plato's and Kant's idea that the experience of musical beauty is universally shared [15,44]. This runs counter, however, to findings from previous studies that musicians tend to experience heightened aesthetic experiences from music [76–79] and that there are different aesthetic and emotional responses to music across different personality traits [54,74,79,91]. The discrepancy between our findings and these previous studies may derive from the measures taken in this study being different from previous studies: this study examined how the number of aspects participants mentioned differs according to musical expertise and personality traits, rather than the intensity of a given emotion that participants experienced. It is thus still possible that, while listeners experience similar breadth or patterns of emotional responding from musical beauty, the intensity of any given emotional response may vary depending on musical expertise and personality traits.

## Limitations

The findings of this study are subject to certain limitations. Some relate to unexamined factors that may be potential contributors to the experience of musical beauty, such as contextual factors [147,148], and listeners' familiarity and musical preference [149,150]. We suggest these factors could be addressed in future studies.

Other limitations relate to methodological choices. Reliance on descriptive self-report may have been restrictive in capturing a multifaceted phenomenon such as musical beauty. More indirect or implicit investigation (such as measuring physiological responses while listening) could provide a valuable complement. Similarly, the within-subjects design we employed, in which the same participants answered the questions regarding both beautiful and not beautiful music, may have artificially sharpened contrasts between responses—a between-subjects design could mitigate this. More specifically, the use of the phrase "listen to often" only in not beautiful block in the questionnaire may have introduced differences in familiarity between beautiful and not beautiful music.

Finally, although the specific thoughts and metacognitive reflections associated with the experience of musical beauty are of considerable interest, they were beyond the scope of our study. These thus remain an area future research could fruitfully explore. Nevertheless, this study offers a rich and compelling insight into listeners' lived experience of musical beauty and the important role it plays in their everyday lives.

## Conclusion

In sum, this study demonstrates that listeners characterise their experience of musical beauty in terms of the features of music and the cognitive and aesthetic processing listeners bring to the experience. We found that the emotions and the psychological impact that come out of experiencing musical beauty—such as being moved, experiencing a feeling of one-ness or of being connected, and gaining emotional support—can play a profound and significant role in individuals' lives. Musical beauty seems to offer the listener not only an opportunity to appreciate music's form and emotional expression, but also consolation and support in challenging times. With regard to individual differences, musical expertise and personality traits can influence the beauty experience; both in terms of the breadth of features that are recognised as important for musical beauty and in terms of the breadth of impact it has on people's musical activities and musical being.

## Supporting information

**S1 File. The prevalence of codes.**
(DOCX)

**S1 Appendix. The list of pieces the participants reported as beautiful and not beautiful.**
(DOCX)

## Acknowledgments

We would like to thank Safiyyah Nawaz, Matilda Leung, and Olivia Tobler for their assistance with genre classifications of musical pieces, and Valentina Camicia for helping with qualitative data analysis.

## Author contributions

**Conceptualization:** Yuko Arthurs, Diana Omigie.

**Data curation:** Yuko Arthurs.

**Formal analysis:** Yuko Arthurs, Eve Merlini, Diana Omigie.

**Funding acquisition:** Yuko Arthurs.

**Methodology:** Yuko Arthurs, Diana Omigie.

**Supervision:** Diana Omigie.

**Visualization:** Yuko Arthurs.

**Writing – original draft:** Yuko Arthurs, Diana Omigie.

**Writing – review & editing:** Yuko Arthurs, Diana Omigie.

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
