## [Decision Letter · Decision Letter 0]

26 May 2025

PONE-D-25-24652Unpacking musical beauty: sound, emotion, and impact differences across expertise and personalityPLOS ONE

Dear Dr. Arthurs,

Thank you for submitting your manuscript to PLOS ONE. After careful consideration, we feel that it has merit but does not fully meet PLOS ONE’s publication criteria as it currently stands. Therefore, we invite you to submit a revised version of the manuscript that addresses the points raised during the review process.

We look forward to receiving your revised manuscript.

Kind regards,

Andrea Schiavio

Academic Editor

PLOS ONE

Reviewers' comments:

Reviewer's Responses to Questions

**Comments to the Author**

1. Is the manuscript technically sound, and do the data support the conclusions?

Reviewer #1: Partly

Reviewer #2: Yes

2. Has the statistical analysis been performed appropriately and rigorously? 

Reviewer #1: Yes

Reviewer #2: Yes

3. Have the authors made all data underlying the findings in their manuscript fully available?

Reviewer #1: Yes

Reviewer #2: Yes

4. Is the manuscript presented in an intelligible fashion and written in standard English?

Reviewer #1: Yes

Reviewer #2: Yes

5. Review Comments to the Author

Reviewer #1: I would like to thank author(s) for the opportunity to review this manuscript. The paper addresses an interesting topic.

This study examined how individuals experience musical beauty by comparing self-selected beautiful and non-beautiful pieces. Analysis of responses revealed that beautiful music is characterized by gentle acoustic features and evokes emotions, such as calmness and pleasant sadness. Beauty evoked by music also provides emotional support and inspiration. Listeners with highly openness were especially sensitive to musical aesthetics, highlighting the role of personality in shaping beauty perception.

However, several points would benefit from further clarification or revision to strengthen the overall quality and impact of the manuscript.

Three points are particularly concerned.

1. The novelty of the research has not been shown.

2. The introduction and discussion do not explain sufficiently most relevant studies that are similar to this study. In the introduction and discussion, although extensive references are provided, they do not appear to be organically integrated with the aim and interpretation of results of this study.

3. Redundancy of descriptions and obscurity of the definition of beauty

The results are presented in a lengthy manner because the elements obtained from the participants' responses are listed in order, which can make them appear redundant. The results are explained using various insights such as emotions and nostalgia, but it is somewhat unclear whether they can truly be explained as “beauty.”

For details, please see below.

Comment#1: Introduction

p.3, l.49-52: This is an overstate, given many psychological studies on music and beauty. To avoid misunderstanding by readers, a more modest expression should be used. In particular, the following papers appear to be related to this study. Fleckenstein et al. (2025) is particularly similar in terms of methodology. This study should discuss previous studies sufficiently and clarify what is known and what is unknown in order to describe the research gap. This will make it easier for readers to understand the significance of this study.

Furthermore, it should be comparable to the latest research by Brattico et al. (2025) using listening experiments. If this study delves into subjective perceptions of beauty using a method different from listening experiments, explaining the similarities and differences with actual listening will help readers gain a deeper understanding.

Fleckenstein, A. M., Vuoskoski, J. K., & Dibben, N. (2025). Understanding Musical Beauty. Empirical Studies of the Arts, 43(1), 505-523.

Brattico, E., Brusa, A., Dietz, M., Jacobsen, T., Fernandes, H. M., Gaggero, G., .. & Proverbio, A. M. (2025). Beauty and the brain–Investigating the neural and musical attributes of beauty during naturalistic music listening. Neuroscience, 567, 308-325.

Comment#2: Introduction Features of music perceived as beautiful

p.4, l.75-76 ‘chills, being moved and the feeling of awe’

It is necessary to explain to readers that chills or being moved are appropriate examples of aesthetic experiences. In many cases, these are often considered in relation to emotion. To avoid such confusion, it can be better to describe (or operationally define) what constitutes an aesthetic experience.

Comment #3: Introduction

p.4, l.79-82: Prior research should be explained thoroughly. Rather than just referring to Omigie et al. [29] in detail, the author(s) should summarize all the studies mentioned in this section concisely, explaining what has been clarified and what has not been clarified in the previous studies cited.

Comment#4: Introduction

p.5, l.95-97: Adding a clear explanation of the relationship between beauty and processing fluency and musical knowledge will make it easier for readers to understand.

Comment #5: Feelings and emotion when experiencing beauty in music

p.6, l.122-123: ‘That perceived beauty and sadness in music are positively correlated [50,52]…’ These studies suggest that sad music can induce positive emotions. If so, simply saying “sadness” is insufficient, and a more specific explanation is preferable.

Comment #6: Individual differences: musical training and personality

p.8, l.174-177: This assumption is probably too definitive. Musicians are expected to be more involved in music. In other words, they are likely to spend more time engaging in music and thinking about it. This will increase their exposure to musical beauty.

Comment #7: Individual differences: musical training and personality

p.9, l.198-200: Based on the previous research, it is necessary to discuss what further consideration is required. This is because the previous research summarized in this paragraph appears to have already been considered to a certain extent.

Comment #8: The present study

p.9, l.203-205: It is difficult to understand what the research gap is. It can be easier for readers to understand if succinct summary is again here.

Comment #9: The present study

p.10, l. 221-223: ‘As for emotion and impact, text responses allow the reporting of rich and diverse responses that might not be possible if participants were provided only with pre-defined emotion labels and statements, e.g., [100,101].’

Specifically, this part should describe the limitations of the methods used in previous studies and how the methods used in this study can overcome those limitations. The same applies to text analysis. The weaknesses of text analysis and the advantages of the methods used in this study should be mentioned.

Comment #10: Participants

p.11, l.242: What is ‘Prolific’?

Comment #11: Survey

p.12, l.260-263:‘“Could you tell us the details of three pieces of music which you find to be very beautiful?” for the beautiful pieces block, and “Now, could you tell us the details of three pieces of music you listen to often but which you wouldn’t describe as beautiful?” for the not beautiful block.’

The phrase “listen to it often” is included when asking about songs that are not beautiful, but why is that? Given such frequencies of listening, wouldn't other factors such as familiarity are related to? Also, why didn't author(s) ask what “beautiful songs they listen to often” are?

Comment #12: Analysis Text Responses to open-ended questions

p.15, l.319-325: Because coding greatly influences the results of this study, it is necessary to mention the reliability of coding. For example, indicators such as the kappa coefficient would be useful.

Comment #13: Results and Findings Pieces volunteered by participants.

p.17, l.358-374: It can be reasonable to assume that experiencing beauty has a significant influence on human behavior. However, if that is the case, why is it that many songs considered beautiful are classical music, which people do not listen to often? Conversely, why is rock music, which is considered less beautiful, listened to?

Comment #14: Table 1. Examples of Responses from Feature Questions.

p.l8-20: Regarding Table 1, why were so many negative responses regarding music that is not beautiful obtained? The question asks about music that is “often listened to but not beautiful.” These responses appear to be impressions of music that the respondents do not like.

Why do respondents “often listen to” songs with such negative characteristics?

Do these responses indicate a gap between beauty and preference? Or did the participants list songs that are “generally considered beautiful”?

Comment #15: Discussion

p.37, l.652-660: The introduction did not clearly explain the research gap, making it difficult for readers to understand what new findings this study revealed. The introduction should clearly state the research gap and then summarize the main findings of this study in a concise manner at the beginning of the discussion in response to that gap.

Comment #16: Discussion

p.37, l.652-660: In the discussion, it is important to clearly state what new findings were revealed in this study that were not previously identified in the literature.

While the author(s) argue that the results of this study are consistent with many previous studies, they do not clearly state the novelty of this study. To help readers understand the novelty of this study, it is important to clearly explain what is new about this study.

Comment #17: Discussion

p.37, l.652-660: In the discussion, “beauty” and “preference” seem to be confused. Does the content that the author(s) are discussing as ‘beautiful’ music also apply to music that is “preferable but not beautiful”? Judging from the participants' responses, beauty seems to be associated with positive contents and, as a result, with preference. Therefore, analyzing these responses seems to be discussing beauty and the positive contents and preferences it contains as a single entity.

Indeed, it is possible to find something beautiful but not like it. This phenomenon appears to be similar to the distinction between “perceived emotion” and “felt emotion” often discussed in terms of musical emotions. Does beauty not have such a state? In any case, careful discussion is necessary about what the participants' responses actually represent.

Comment #18: Discussion

p.37, l.652-660: The analysis and results are wide-ranging and contain many contents related to beauty, making it difficult to grasp. It can be easier for readers to understand if the results of “beauty” obtained in this study could be modeled or diagrammed as a figure.

Comment #19: Discussion

p.37, l.662-663: The sub-heading is long and vague, so it should be made more concise and to the point.

Comment #20: Discussion Beautiful music is characterised by its intrinsic features, cognitive

structural factors and listeners’ application of aesthetic criteria

p.38, l.682-683: ‘while Complexity level is reminiscent of the inverted-U [92,93] and processing fluency theories [32].’

Please explain in detail. Have these theories been used to explain beauty in previous studies? Otherwise, is this a new finding of this study?

Comment #21: Discussion The affective profile of beautiful music: feeling calm, pleasurably sad, being moved and feeling emotionally supported

p.40, l.715-772: Because discussion about beauty and preferences is lack, it should be discussed.

Comment #22: Discussion The affective profile of beautiful music: feeling calm, pleasurably sad, being moved and feeling emotionally supported

p.42, l.760-761: ‘There are arguably two reasons why the pieces that listeners find beautiful might help them to deal with emotional difficulties.’

Do these conclusions apply to music that is “not beautiful but preferable”? Ultimately, as mentioned earlier, it is necessary to pay attention to what the participants' responses represent in order to draw conclusions.

Comment #23: Discussion Beautiful music impacts mood, evokes the past, motivates musical

activities, but is not used to shape identity

p.43, l.782-784: ‘Another impact many mentioned is that listening to beautiful pieces was able to help them retrieve past events, people or feelings in memories, as discussed in the wide literature on music and autobiographical memories, e.g., [127].’

Isn't this a case of reverse causality? Could it be that we find certain songs beautiful because they are associated with specific memories? Or is it that beautiful songs are more likely to be associated with memories? In any case, evidence is needed to support this argument.

Comment #24: Discussion Beautiful music impacts mood, evokes the past, motivates musical

activities, but is not used to shape identity

p.43, l.792-796: ‘Responses revealed that while listening to both beautiful and not beautiful pieces stimulate the listener’s musical curiosity and intelligence (such as the desire to understand the structure of the piece, enjoy its complexity, or discover novel musical expressions), listening to beautiful pieces tended to give rise to the desires to engage in further musical activities.’

This part seems to contradict the subsequent discussion. Ultimately, does beauty promote motivation, or is there some other mediating factor?

Comment #25: Discussion Beautiful music impacts mood, evokes the past, motivates musical

activities, but is not used to shape identity

p.44, l.802-804: ‘Participants also reported that listening to not beautiful pieces influenced their musical preference for particular genres and helps shape their musical identity and identity in a broader sense, in line with previous studies [130–132].’

Why do not beautiful songs contribute to identity formation? How have previous studies discussed such tendency?

Comment #26: Discussion Individual differences in musical beauty experience

p.44, l.811-834: The reasons for the results differing from those of previous studies should be discussed in detail. Otherwise, it will be difficult for readers to understand what these results represent.

Comment #27: Discussion Individual differences in musical beauty experience

p.44, l.822-825: ‘For instance, professional musicians tended to mention Intrinsic features of sound and music (Feature) in beautiful music more frequently, reflecting musicians’ finer perceptual and cognitive musical abilities, e.g., [133], and their greater ability to explain music and their musical experiences using technical terms, e.g., [134].’

Does this tendency indicate that musicians and non-musicians have different perceptions of what constitutes “beauty”? If so, then the results obtained from the responses may simply reflect individual differences in the interpretation of the word “beauty,” rather than a specific ability to perceive music as beautiful.

Comment #28: Discussion

p.45, l.842-848: ‘Musical beauty seems to offer the listener not only an opportunity to appreciate music’s form, emotional expression and originality, but can also offer consolation and support in challenging times. Interestingly, while some intuitive trends were observed with regard to how individual differences influence the experience of musical beauty, our results were largely in keeping with Plato’s and Kant’s idea, supported by recent empirical work [11] that the experience of musical beauty (specifically features perceived as important, emotional responding, and psychological impact) is largely universal.’

Since the respondents' comments were so diverse, isn't it natural that there would be individual differences? Just because no individual differences were observed in terms of personality trait does not mean that it is an exaggeration to generalize beauty.

Reviewer #2: Thank you for the opportunity to review the manuscript entitled "Unpacking musical beauty: sound, emotion, and impact differences across expertise and personality" (Manuscript ID: PONE-D-25-24652), submitted to PLOS ONE.

This manuscript uses a mixed-methods approach to explore the concept of musical beauty, with a sample of 81 adult participants. Participants were asked to provide qualitative reflections on musical features, emotional responses and the long-term impact of pieces they considered beautiful, as opposed to pieces they liked but would not describe as beautiful. Through thematic analysis, the authors identified recurring themes associated with the perception of musical beauty. A subsequent quantitative analysis used Poisson regression to examine whether differences in musical training and personality traits predicted how often participants referred to specific themes when reflecting on beautiful versus non-beautiful music.

General comments:

Overall, I think the manuscript is well written, clearly organized, and engaging. I appreciated the way in which the authors combined qualitative insights with quantitative analysis, and really enjoyed reading the manuscript. In my opinion, the manuscript is almost ready for publication. I have provided some suggestions for minor revisions in the detailed comments below. which I hope the authors will find useful.

Detailed comments:

Abstract:

The abstract is clear and well-written. It is especially helpful that it immediately summarizes the key characteristics that according to the qualitative results distinguish music that is perceived as beautiful from music that is merely liked. To improve clarity, the authors could provide brief, concrete examples of the high-level features mentioned, such as specifying “a balanced level of complexity” rather than the more general “complexity level”.

Introduction:

- The introduction is well-structured and provides a comprehensive overview of previous research into the concept of musical beauty. One area that could benefit from further clarification in my view is the distinction between beauty, aesthetic emotions (such as awe) and cognitive judgements about music. Since musical beauty is conceptualized differently in different studies (sometimes more as an aesthetic emotion and sometimes more as a cognitive judgement) these different perspectives could be introduced more explicitly at an earlier stage.

- Prior to discussing previous research on the characteristics of beautiful music, the authors could introduce the three main focus areas of the study (musical features, emotional responses and long-term impact) more clearly. Since other dimensions, such as cognitive responses (e.g. thoughts evoked by the music), could also be relevant, providing a brief rationale for why exactly these three areas were chosen would strengthen the framing of the study.

- In line 61, beauty is described as a key factor in enabling positive aesthetic judgement. However, this may imply a directional relationship that is not clearly established, as the perception of beauty could also follow positive aesthetic judgement. Could you perhaps clarify this, or formulate both directions?

- Line 86: It would be helpful to explain why only tension and energy are mentioned in the context of the Omigie et al. study. Were these the most commonly reported emotions in this study or the only emotions contributing to different emotional profiles of beautiful vs. not-beautiful music pieces?

- Line 96 – “Perceived beauty has been argued to be influenced by the ease with which a piece a piece can be cognitively processed” Here the direction of the effect could be clarified. Is beauty associated with greater or lesser processing ease?

- When discussing the possible influence of musical expertise on the perception of beauty, it would be beneficial to introduce the concept of knowledge-based emotions at an earlier stage in the manuscript. Currently, this concept is only briefly mentioned in the discussion.

Method:

- I recommend adding the age range of the participants to the sample description. If available, it would also be helpful to include more demographic information, such as the participants' country of residence and educational background. It would also be useful to state whether any eligibility criteria or filters were applied during the recruitment process. Since cultural background may influence perceptions of musical beauty, including these details would provide important context for interpreting the findings.

- Line 303 – “however, these measurements were not included in the current analysis.” Consider rephrasing this sentence to make it clear that only the EBS was excluded from the analysis (and not both EBS and TIPI)

Analysis:

- The subheadings in the analysis section could be more specific. Consider using headings such as “Qualitative analysis of open-ended responses” and “Quantitative analysis of the influence of musicianship and personality traits”.

- Thematic analysis: The description of the coding process could be clearer. To me, it was not entirely clear whether only Author 2 initially coded all responses and then discussed the codes with others, or whether multiple authors independently coded the data. If multiple coders rated the same responses, it would be important to report an inter-rater reliability metric, such as Fleiss’ Kappa.

- Line 350 – “Poisson mixed effect models were run to examine the influence of music type and listener individual differences on these dependent variables of interest”: It took me some time to understand that the dependent variables were the themes found within the three core topics of features, emotions and impacts. This could be stated more clearly.

- Line 351-356 “with participant ID and pieces included as random effects”: From the phrasing of the open-ended questions, it appears that participants were asked within the same question to describe features (in the other questions emotions or impacts) across three selected musical pieces, as well as more generally for music they consider beautiful or not beautiful. If so, it may have been difficult to distinguish whether the responses referred to the specific pieces or the category as a whole. This raises the question of whether it would have been more appropriate to count themes at the condition level (beautiful vs. non-beautiful) rather than for each individual piece. Aggregating theme counts by condition could also reduce model complexity of the Poisson regressions. However, I may have misunderstood this part of the procedure. If so, a clearer explanation in the manuscript would be helpful.

Results:

- Figure 1/Table 1: To me, it was not immediately obvious that the quotes in Table 1 are not organized by the individual codes listed in Figure 1. This could be made clearer. One way to achieve this would be to present one representative quote per code, or alternatively, to label each quote with the corresponding code name.

- Table 5 & 6: Given the relatively small sample size, using Poisson regression models with 15 predictors could be too ambitious and risk overfitting. There may be alternative modelling strategies worth exploring. For instance, computing difference scores (i.e., the frequency of the theme being mentioned in the beautiful music condition minus the frequency in non-beautiful music condition) could streamline the analysis by eliminating the need for interaction terms. But this approach would likely require a linear regression model instead of a Poisson model, depending on the distribution of the difference scores. Another option would be to analyze theme frequencies only for the beautiful music condition, reducing model complexity and possibly yielding clearer insights. Although I am not a specialist in mixed Poisson regression, I would like to offer these suggestions as possible alternatives that might help to address the models' complexity relative to the sample size.

Discussion:

- One area that could be expanded upon is the comparison of the present results with those of Fleckenstein et al. (2025). While the authors reference this study when discussing specific codes or themes, a more direct comparison of the relative prominence of themes identified in both studies would be valuable.

- Line 752-759: The link between musical beauty and emotion regulation is very interesting. Perhaps the authors could elaborate more on this point. For instance, different types of music (beautiful versus not beautiful) might serve distinct functions in mood regulation. High-energy, less beautiful music might be more suitable for emotional release or catharsis (e.g. discharging tension or negative emotions), whereas low-energy, beautiful music may be better suited to emotional reflection or finding emotional comfort.

6. PLOS authors have the option to publish the peer review history of their article (what does this mean? ). If published, this will include your full peer review and any attached files.

**Do you want your identity to be public for this peer review?** For information about this choice, including consent withdrawal, please see our Privacy Policy .

Reviewer #1: No

Reviewer #2: No

---

## [Author Response · Author response to Decision Letter 1]

20 Jul 2025

We would like to thank the reviewers for their valuable and constructive comments and helpful suggestions. We truly appreciate the time and effort invested in reviewing this manuscript. We have carefully considered all suggestions and comments, and revised the manuscript accordingly. The detailed responses to each comments are provided below.

Reviewer #1:

I would like to thank author(s) for the opportunity to review this manuscript. The paper addresses an interesting topic.

This study examined how individuals experience musical beauty by comparing self-selected beautiful and non-beautiful pieces. Analysis of responses revealed that beautiful music is characterized by gentle acoustic features and evokes emotions, such as calmness and pleasant sadness. Beauty evoked by music also provides emotional support and inspiration. Listeners with highly openness were especially sensitive to musical aesthetics, highlighting the role of personality in shaping beauty perception.

However, several points would benefit from further clarification or revision to strengthen the overall quality and impact of the manuscript.

Three points are particularly concerned.

1. The novelty of the research has not been shown.

2. The introduction and discussion do not explain sufficiently most relevant studies that are similar to this study. In the introduction and discussion, although extensive references are provided, they do not appear to be organically integrated with the aim and interpretation of results of this study.

3. Redundancy of descriptions and obscurity of the definition of beauty

The results are presented in a lengthy manner because the elements obtained from the participants' responses are listed in order, which can make them appear redundant. The results are explained using various insights such as emotions and nostalgia, but it is somewhat unclear whether they can truly be explained as “beauty.”

For details, please see below.

Thank you very much for pointing out these three points.

1. We have now worked carefully to clarify the study’s novelty throughout the manuscript.

2. All reviewed literature (apart from the earliest context-setting opening comments) now directly and explicitly motivate the gaps we have identified in the introduction, and we now very explicitly contextualize all findings in our Discussion section.

3. We have carefully revisited the results to make them more concise and logically presented.

The details of changes made can be seen under each comment.

Comment#1: Introduction

p.3, l.49-52: This is an overstate, given many psychological studies on music and beauty. To avoid misunderstanding by readers, a more modest expression should be used. In particular, the following papers appear to be related to this study. Fleckenstein et al. (2025) is particularly similar in terms of methodology. This study should discuss previous studies sufficiently and clarify what is known and what is unknown in order to describe the research gap. This will make it easier for readers to understand the significance of this study.

Furthermore, it should be comparable to the latest research by Brattico et al. (2025) using listening experiments. If this study delves into subjective perceptions of beauty using a method different from listening experiments, explaining the similarities and differences with actual listening will help readers gain a deeper understanding.

Fleckenstein, A. M., Vuoskoski, J. K., & Dibben, N. (2025). Understanding Musical Beauty. Empirical Studies of the Arts, 43(1), 505-523.

Brattico, E., Brusa, A., Dietz, M., Jacobsen, T., Fernandes, H. M., Gaggero, G., .. & Proverbio, A. M. (2025). Beauty and the brain–Investigating the neural and musical attributes of beauty during naturalistic music listening. Neuroscience, 567, 308-325.

Thank you for your comments. We agree that we may have misrepresented the volume of recent work that has been done in this area. We also agree that these two studies may be two of the more similar papers to ours with respect to the aims and the questions addressed. We now describe them in the necessary detail in the introduction, clarifying where remaining research gaps lie and how our study addresses these research gaps. Importantly, we now also provide more details about the many other relevant studies: this, so as to more clearly demonstrate the significance of the current study.

Specifically, firstly, instead of saying ‘at least from a psychological point of view, the experience of musical beauty remains largely unexplored’, we say that:

-Line 51-52, page 3

‘research on musical beauty from a psychological perspective is still relatively limited’

Secondly, we have highlighted Fleckenstein et al. (2024) ‘s paper as below

Line 74, page 4

‘Recent studies have used wide-ranging methods to tackle some of the fundamental questions regarding musical beauty. One study by Fleckenstein et al [12] was particularly valuable in exploring participants’ subjective definition of musical beauty. Their study revealed that participants mainly view musical beauty as both an emotional experience on the one hand and as determined by the properties of an object on the other. The study also identified various factors that contribute to the experience of musical beauty, including participants' felt emotional experiences, the emotions expressed by the music, the timbre of instruments, and the memories listeners associated with the music.’

Thirdly, we have highlighted Brattico et al. (2025) ‘s paper as below

-Line 81, p.4

‘Complementing the qualitative approach taken in that study, Brattico et al [10] investigated the neural correlates of the experience of musical beauty, differentially associating different structures (orbitofrontal activity and bilateral supratemporal activity) with the experience of beauty. That study also revealed differences in sonic features of beautiful and not beautiful sections of music whereby beautiful, compared to not beautiful, sections were rated (by composers) as more tonal, simple, melodic, traditional, calm and sad.’

Comment#2: Introduction Features of music perceived as beautiful

p.4, l.75-76 ‘chills, being moved and the feeling of awe’

It is necessary to explain to readers that chills or being moved are appropriate examples of aesthetic experiences. In many cases, these are often considered in relation to emotion. To avoid such confusion, it can be better to describe (or operationally define) what constitutes an aesthetic experience.

Thank you for this point. However, we have now removed the reference to aesthetic experience in this sentence as we realized it was making the sentence unnecessarily complicated.

We have nevertheless introduced the definition of an aesthetic judgment later on when referring to the literature that uses that term.

-Line 67. page 3

‘An aesthetic judgement is considered an output, along with emotion and preferences, of the processing of artworks and other objects. This form of processing is held to involve perceptual, cognitive, and affective components, e.g., [21,25,26]’

Comment #3: Introduction

p.4, l.79-82: Prior research should be explained thoroughly. Rather than just referring to Omigie et al. [29] in detail, the author(s) should summarize all the studies mentioned in this section concisely, explaining what has been clarified and what has not been clarified in the previous studies cited.

Thank you. We agree. This revised paragraph now concisely summarises many relevant studies and clarifies the research gap.

-Line 108-127, page 5-6

‘The extent to which specific sonic features might be tied to key experiences of music (such as chills, being moved and the feeling of awe) has seen significant interest in the music psychology literature [30–33]. These studies have aligned in highlighting the role of both lower-level musical elements (such as loudness, tempo, and timbre) and higher-order, structural characteristics (such as tonal clarity) in affording these experiences. Regarding musical beauty, consonance—two or more harmoniously sounding concurrent tones—has been traditionally associated with beauty in the context of Western tonal music [34,35]. More recently, Fleckenstein et al.’s [12] participants reported not just harmonic content, but also instrumental timbre and tempo/rhythm/meter as factors contributing to their experience of musical beauty.

Interestingly, studies which have examined sections of pieces considered beautiful by listeners highlight the potential importance of temporal changes in musical elements for the experience of musical beauty. Beautiful sections in music, compared to not beautiful sections have been associated with more tonal, simple, melodic, traditional features [10], raising the possibility that listeners may find these moments more beautiful due to how they contrast with other moments. Similarly, a study by Omigie et al [13] demonstrated that while beautiful passages or sections of music could can vary in characteristics, they are often characterised by change such as a drop in tempo and a reduction in polyphony in some cases, or increases in dynamics, pitch register, major tonality, and harmonic clarity, in others.’

Comment#4: Introduction

p.5, l.95-97: Adding a clear explanation of the relationship between beauty and processing fluency and musical knowledge will make it easier for readers to understand.

Thank you for pointing this out. We have now rephrased this sentence as follows;

-Line 130, page 6

‘Indeed, according to the processing fluency theory [38], perceived beauty may be influenced by the ease with which a piece can be cognitively processed, where the easier it is to process the piece, the more beauty may be experienced.’

Comment #5: Feelings and emotion when experiencing beauty in music

p.6, l.122-123: ‘That perceived beauty and sadness in music are positively correlated [50,52]…’ These studies suggest that sad music can induce positive emotions. If so, simply saying “sadness” is insufficient, and a more specific explanation is preferable.

Apologies that we were not clear. Whilst these studies showed positive relationships between beauty and sadness (hence our statement), we have now revised the sentence to make that clearer, and to emphasise further reasons (including the role of epistemic emotions) why beauty should be considered multifaceted and going beyond positive valance.

-Line 157, page 7

‘This research shows that, alongside positive valence or pleasure, experiencing beauty from artworks is often linked with the feeling of being moved [33,49,50], the feeling of positive awe [32], and the feeling of sadness [51–54]. Further, hinting at the additional role that epistemic emotions may play, Armstrong and Detweiler-Badell [47] argue that the desire to understand a piece and the satisfactory resolution of such desire through apprehending the piece, are essential for experiencing beauty.

The variety of emotions associated with beauty experiences so far suggest that characterising beauty as pleasure alone may not fully capture its multifacetedness, compared to other forms of pleasure [46]. However, in light of ongoing debates about the relevance of distinguishing aesthetic pleasure from other forms [46], systematic research into the emotions that accompany beauty experiences may be considered increasingly necessary.’

Comment #6: Individual differences: musical training and personality

p.8, l.174-177: This assumption is probably too definitive. Musicians are expected to be more involved in music. In other words, they are likely to spend more time engaging in music and thinking about it. This will increase their exposure to musical beauty.

Thank you for this comment. We initially meant to suggest that musicians may have been more impacted by beauty in the past and that is the reason why they choose their profession. However, we agree that the direction of causality could go both ways, since it is also possible that those who choose to be a musician may consequently then experience more impact from music.

We have now clarified these alternative possibilities while pointing out that both would nevertheless be reflected in a positive relationship between the musicianship and reported impact of beautiful music.

-Line 218, page 10.

‘Further, since it has been reported that people who pursue music professionally often had strong emotional and aesthetic experiences of music during childhood [64,65], or a strong sense of calling for music during adolescence [66], one could predict that musicians, more than non-musicians, will report having been more impacted by musical beauty over the course of their lives. Here, it is important to note that being a musician may provide more opportunities to experience the possible impacts of music, and that as such impacts reported this group may be higher for that reason. In either case, a positive relationship between musicianship and breadth or scale of impacts from beautiful music seems a reasonable prediction to make.’

Comment #7: Individual differences: musical training and personality

p.9, l.198-200: Based on the previous research, it is necessary to discuss what further consideration is required. This is because the previous research summarized in this paragraph appears to have already been considered to a certain extent.

Thank you for mentioning this. We have now revised this paragraph to clarify that what further consideration is required;

-Line 247, page 11

‘Taken together, the findings from previous studies point to the possibility that musical beauty may subjectively differ according to the listener’s musical training and personality traits. Musicianship and Openness to experience may influence the ability to discuss features contributing to the experience of music, while Openness to experience and/or Extraversion may lead to more varied experience of emotions in response to music perceived as beautiful. Musicians may also be expected to report a wider range of impacts from music.’

Comment #8: The present study

p.9, l.203-205: It is difficult to understand what the research gap is. It can be easier for readers to understand if succinct summary is again here.

Thank you for your careful review and comment. As requested, we have now made the gap very clear in previous sections throughout the manuscript. This means however that another summary here would be highly redundant.

Comment #9: The present study

p.10, l. 221-223: ‘As for emotion and impact, text responses allow the reporting of rich and diverse responses that might not be possible if participants were provided only with pre-defined emotion labels and statements, e.g., [100,101].’

Specifically, this part should describe the limitations of the methods used in previous studies and how the methods used in this study can overcome those limitations. The same applies to text analysis. The weaknesses of text analysis and the advantages of the methods used in this study should be mentioned.

Thank you for your comments. The limitations of the methods employed in previous studies have now been identified earlier when we describe them and here, we stress the advantages of the methods employed in this study, as follows;

-Line 270, page 12.

Note: “[the advantage]”, ”[the limitation]” and highlighting have been added here for clarification only.

‘Specifically, regarding exploring the features of sound in music associated with beauty, text responses reveal listeners’ psychological impressions of both lower and higher- level features of sound [the advantage] in ways that might not be possible using automated feature extraction techniques [the limitation]. As for emotion and impact, text responses allow the reporting of rich and diverse responses [the advantage] that might not be possible if participants were provided only with pre-defined emotion labels and statements [the limitation], e.g., [100,101].’

Comment #10: Participants

p.11, l.242: What is ‘Prolific’?

We’re sorry that we didn’t provide a clear description. Prolific is an online platform where researchers can recruit participants for their online studies. We have added this information to the manuscript;

-Line 296, page 14.

‘A total of 81 participants.., having been recruited through

---

## [Decision Letter · Decision Letter 1]

7 Sep 2025

PONE-D-25-24652R1Unpacking musical beauty: sound, emotion, and impact differences across expertise and personalityPLOS ONE

Dear Dr. Arthurs,

Thank you for submitting your manuscript to PLOS ONE. After careful consideration, we feel that it has merit but does not fully meet PLOS ONE’s publication criteria as it currently stands. Therefore, we invite you to submit a revised version of the manuscript that addresses the points raised during the review process.

We look forward to receiving your revised manuscript.

Kind regards,

Andrea Schiavio

Academic Editor

PLOS ONE

Journal Requirements:

Reviewers' comments:

Reviewer's Responses to Questions

**Comments to the Author**

1. If the authors have adequately addressed your comments raised in a previous round of review and you feel that this manuscript is now acceptable for publication, you may indicate that here to bypass the “Comments to the Author” section, enter your conflict of interest statement in the “Confidential to Editor” section, and submit your "Accept" recommendation.

Reviewer #1: All comments have been addressed

Reviewer #2: (No Response)

2. Is the manuscript technically sound, and do the data support the conclusions?

Reviewer #1: Yes

Reviewer #2: Yes

3. Has the statistical analysis been performed appropriately and rigorously? 

Reviewer #1: Yes

Reviewer #2: Yes

4. Have the authors made all data underlying the findings in their manuscript fully available?

Reviewer #1: Yes

Reviewer #2: No

5. Is the manuscript presented in an intelligible fashion and written in standard English?

Reviewer #1: Yes

Reviewer #2: Yes

6. Review Comments to the Author

Reviewer #1: I would like to thank author(s) for the opportunity to review this revised manuscript.

The authors have made substantial improvements compared to the initial submission, and the paper is now clearer and more convincing overall. I appreciate the efforts taken to revise the manuscript in response to the earlier comments.

That said, there remain several points that require further attention. These issues are not as extensive as in the first round, but they are nevertheless important for ensuring the clarity, rigor, and overall contribution of the work. I outline these specific concerns in the detailed comments below.

Comment #11: Survey

p.12, l.260-263:‘“Could you tell us the details of three pieces of music which you find to be very beautiful?” for the beautiful pieces block, and “Now, could you tell us the details of three pieces of music you listen to often but which you wouldn’t describe as beautiful?” for the not beautiful block.’

The phrase “listen to it often” is included when asking about songs that are not beautiful, but why is that? Given such frequencies of listening, wouldn't other factors such as familiarity are related to? Also, why didn't author(s) ask what “beautiful songs they listen to often” are?

Response:

We included the phrase ‘listen to often’ for not beautiful pieces to prevent participants from conflating not beautiful music with disliked music. We wanted to make sure that participants provide not-beautiful pieces they choose to listen to as they choose to listen to beautiful pieces.

However, we agree that it may have helped reduce any effect of familiarity if we had requested that participants report beautiful music they listen to often to match our control condition. We now mention this as a limitation.

-Line 1044, page 64.

‘Additionally, our study did not examine how familiarity or preference for the pieces listed by participants might have influenced their experience of musical beauty. Future studies could impose more control to ensure that participants’ level of familiarity and liking are maximally comparable between beautiful and not beautiful pieces.’

Comment #11: Round2

The implications of this instruction should be considered in interpreting the results. The author(s) mentioned the association between beauty and processing fluency. Since familiarity correlates with processing fluency, the instruction “listen to it often” itself may be a factor related to beauty. If control for factors were to be exercised, why wasn't the instruction phrased as “beautiful music you listen to often”? Although beauty and liking are not entirely independent, the results derived from this instruction warrant more careful discussion. Among the two factors—beautiful/not beautiful, listen often/don't listen often—wouldn't “not beautiful and not listened to it often” be considered the least beautiful music?

Comment #13: Results and Findings Pieces volunteered by participants.

p.17, l.358-374: It can be reasonable to assume that experiencing beauty has a significant influence on human behavior. However, if that is the case, why is it that many songs considered beautiful are classical music, which people do not listen to often? Conversely, why is rock music, which is considered less beautiful, listened to?

Response:

Thanks for this interesting comment. We would argue, however, that even if beautiful music was listened to less frequently than music not found beautiful it would still have the ability to influence behaviour as much if not more. For instance, it is possible that beautiful music (which can often be classical music) is listened to more attentively than non beautiful music and as such has a more significant emotional impact. We, however, refrain from adding this speculation to the manuscript in the interest of brevity.

Comment #13: Round2

This explanation requires further elaboration. One motivation for playing an instrument is a strong emotional experience, and certainly, the beauty of music is involved in such experiences. However, what is crucial here is the emotional experience itself—is beauty a necessary condition? Ultimately, as with the previous comment, beauty and personal preference are intertwined, so it is necessary to clarify these concepts before proceeding with the discussion.

Comment #14: Table 1. Examples of Responses from Feature Questions.p.l8-20: Regarding Table 1, why were so many negative responses regarding music that is not beautiful obtained? The question asks about music that is “often listened to but not beautiful.” These responses appear to be impressions of music that the respondents do not like.

Why do respondents “often listen to” songs with such negative characteristics?

Do these responses indicate a gap between beauty and preference? Or did the participants list songs that are “generally considered beautiful”?

Response:

Thank you for this question. We want to stress that negative feelings here include feelings like agitation and restlessness which while not positive, might be aesthetically valuable to a listener. In that vein, the occurrence of negative feelings may not be so surprising.

Comment #14: Round 2

Please incorporate the content here into the body text.

Reviewer #2: Dear authors,

Thank you for your additional work on the manuscript entitled “Unpacking musical beauty: sound, emotion, and impact differences across expertise and personality" (Manuscript ID: PONE-D-25-24652). Overall, I believe the manuscript has improved considerably, and I do not have any major concerns at this stage.

However, I have a few minor suggestions that may help strengthen the final version.

1. Language use & style: The manuscript is quite lengthy, and some sections are repetitive. For example, the sentence “In addition to lacking a clear definition, also frequently debated is how beauty should be conceptualised.” repeats the same idea. Similarly, in this passage the word "recognized" is repeated in consecutive sentences: "Here, it is recognised that it may be a positive experience for the listener when a piece of music unfolds as they anticipated [40,41]. However, it is also increasingly recognised that a degree of unexpectedness, surprise, or novelty in music can lead to greater feelings of pleasure.” These are just a couple of examples, but there are other similar instances throughout the text. I recommend carefully reading through the text with an eye toward tightening the language and reducing redundancy. You might also consider having a native English speaker review the manuscript to improve clarity and flow.

2. Long-term influences: The introduction mentions the long-term impact of experiencing beauty in music as a novel feature of this study. However, this aspect is not revisited in depth later in the manuscript. It is also unclear from the questionnaire whether participants were prompted to reflect on short-term or long-term effects. To clarify, it would be helpful to state in the introduction that the study explores both immediate and potentially lasting impacts. Consider referencing this again in the results and/or discussion.

3. Discussion: I felt that the new discussion section headings were too long and complex. To improve readability and structure, I suggest simplifying them. For example: “Intrinsic, structural, and aesthetic elements of beautiful music”, “Emotional responses to beautiful music”, “Short- and long-term impacts of beautiful music”, “Individual differences in experiences of beauty”

4. Figure 8 & Abstract: For Figure 8, consider providing a more balanced level of detail across all three sections. In the section "What beautiful music makes people feel" you list specific emotional responses reported by participants. However, in "What makes music beautiful", you only list broader categories (e.g., timbre, melody) without providing concrete examples of what was perceived as beautiful. Similarily, the abstract mentions emotional and impact-related findings but does not touch on the specific musical features that participants associated with beauty. Since many readers may only read the abstract or jump directly to the discussion and this nice figure, clearly and concisely summarizing all key findings in both Figure 8 and the abstract would strengthen the accessibility and impact of your work.

5. Limitations: Another potential limitation you might mention is the within-subjects design. Asking participants to reflect on beautiful and non-beautiful music may have encouraged them to emphasize the differences between the two, which could have influenced their responses. A future between-subjects design could have yielded different results and potentially reduced this contrast effect.

7. PLOS authors have the option to publish the peer review history of their article (what does this mean? ). If published, this will include your full peer review and any attached files.

**Do you want your identity to be public for this peer review?** For information about this choice, including consent withdrawal, please see our Privacy Policy .

Reviewer #1: No

Reviewer #2: No

---

## [Author Response · Author response to Decision Letter 2]

4 Oct 2025

Response_Letter2

Dear reviewers,

Thank you very much for the time taken to read our manuscript again. We appreciate the reviewers’ suggestion, and the manuscript has been revised accordingly, with modifications that enhance readability. Our detailed responses to each comment are provided below.

Reviewer #1:

Comment #11: Survey

p.12, l.260-263:‘“Could you tell us the details of three pieces of music which you find to be very beautiful?” for the beautiful pieces block, and “Now, could you tell us the details of three pieces of music you listen to often but which you wouldn’t describe as beautiful?” for the not beautiful block.’

The phrase “listen to it often” is included when asking about songs that are not beautiful, but why is that? Given such frequencies of listening, wouldn't other factors such as familiarity are related to? Also, why didn't author(s) ask what “beautiful songs they listen to often” are?

Response:

We included the phrase ‘listen to often’ for not beautiful pieces to prevent participants from conflating not beautiful music with disliked music. We wanted to make sure that participants provide not-beautiful pieces they choose to listen to as they choose to listen to beautiful pieces.

However, we agree that it may have helped reduce any effect of familiarity if we had requested that participants report beautiful music they listen to often to match our control condition. We now mention this as a limitation.

-Line 1044, page 64.

‘Additionally, our study did not examine how familiarity or preference for the pieces listed by participants might have influenced their experience of musical beauty. Future studies could impose more control to ensure that participants’ level of familiarity and liking are maximally comparable between beautiful and not beautiful pieces.’

Comment #11: Round2

The implications of this instruction should be considered in interpreting the results. The author(s) mentioned the association between beauty and processing fluency. Since familiarity correlates with processing fluency, the instruction “listen to it often” itself may be a factor related to beauty. If control for factors were to be exercised, why wasn't the instruction phrased as “beautiful music you listen to often”? Although beauty and liking are not entirely independent, the results derived from this instruction warrant more careful discussion. Among the two factors—beautiful/not beautiful, listen often/don't listen often—wouldn't “not beautiful and not listened to it often” be considered the least beautiful music?

***We would not want to argue that "Not beautiful and not listened to it often” music is the least beautiful because that would be relying heavily on the processing fluency account of beauty, when in fact we are keen to hear from participants themselves what music they find beautiful and not beautiful. However, we agree that in accordance with the processing fluency theory, the more music is listened to, the more it may be experienced as beautiful and less listened to the more it may be considered not beautiful. We also now see that, therefore, when we specify non beautiful pieces be pieces they listen to often, we may be leading them to mention pieces that while not beautiful, may still have decent levels of processing fluency for the listener.

As previously mentioned, we wanted to ensure that our control conditions included pieces of music that people wanted to and are likely to be hearing in everyday life. We, in other words, wanted to provide a naturalistic comparator group for our beautiful music condition i.e., we did not want to compare our beautiful music to the ‘ugliest’ music people can think of which they and other may not listen to in everyday life. We realise that using the word listen to often in not beautiful condition may introduce potential differences in the frequencies of music in the two conditions, but think that the benefits of our approach may outweigh negative in this case. We add this rational and justification even more clearly in our manuscript whilst leaving in the sections where we describe our limitations.

Lines 321-324, p. 15: Procedure

“We specified “listen to often’ when referring to not beautiful music to ensure that participants did not conflate ‘not beautiful’ with ‘disliked’ or ‘ugly’ music they would normally try to avoid, but rather considered not beautiful music that they nevertheless engage with in everyday life.”

Lines 1027-29, p.63: Limitations

“the use of the phrase “listen to often” only in not beautiful block in the questionnaire may have introduced differences in familiarity between beautiful and not beautiful music.”

Comment #13: Results and Findings Pieces volunteered by participants.

p.17, l.358-374: It can be reasonable to assume that experiencing beauty has a significant influence on human behavior. However, if that is the case, why is it that many songs considered beautiful are classical music, which people do not listen to often? Conversely, why is rock music, which is considered less beautiful, listened to?

Response:

Thanks for this interesting comment. We would argue, however, that even if beautiful music was listened to less frequently than music not found beautiful, it would still have the ability to influence behaviour as much if not more. For instance, it is possible that beautiful music (which can often be classical music) is listened to more attentively than non beautiful music and as such has a more significant emotional impact. We, however, refrain from adding this speculation to the manuscript in the interest of brevity.

Comment #13: Round2

This explanation requires further elaboration. One motivation for playing an instrument is a strong emotional experience, and certainly, the beauty of music is involved in such experiences. However, what is crucial here is the emotional experience itself—is beauty a necessary condition? Ultimately, as with the previous comment, beauty and personal preference are intertwined, so it is necessary to clarify these concepts before proceeding with the discussion.

***The section that the reviewer is referring to describe the frequencies with which beautiful and not beautiful music come from different genres. We think the reviewer, however, is keen to understand why music like rock, even if it’s not beautiful, may lead people to want to play music (that is why not beautiful music may impact behaviour). We think this is an interesting question, and to be clear we think, especially based on our results, that music does not have to be beautiful to impact behaviour. In other words, beauty is not a necessarily condition to impact behaviour and that preferred music can have this impact too. We point the reader to where this is clear from our manuscript.

For example,

Lines 683-695, p.45

“Memory and the Self

Participants’ comments demonstrated that while both beautiful and not beautiful music helped them connect with their lives, not beautiful music showed a higher tendency for codes related to everyday use and identity (57.14% to 100%). Participants described how both beautiful and not beautiful lead them to relive memories (I-“Remembering the Past”-B11, I-“Remembering the Past”-NB11), of past situations in which they listened to and/or performed the piece (I-“Remembering the Past”-B11), and with associated memories that were not always positive (I-“Remembering the Past”-B12, I-“Remembering the Past”-NB12). However, not beautiful music especially was described as serving as a practical aid to daily life activities, from housework, to the daily commute, and even to sleep (I-“Practical Chore”-B13, I-“Practical Chore”-NB13). Similarly, not beautiful music exclusively was described as having shaped the participants’ identity or personality (I-“Life Soundtrack”-NB14, Beautiful = 0%, Not Beautiful = 100%).”

Lines 704 – 712, p. 48

“Finally, it is important to note that participants reported not beautiful music especially (Beautiful = 35.71%, Not Beautiful = 64.29%) as having stimulated their musical curiosity (I-“Stimulating Curiosity”-NB18) and having widened their repertoire or interest (I-“Stimulating Curiosity”-B18). It is also notable that (similar to beautiful music never being reported as having shaped personal identity (see Memory and the self)), there was a greater tendency for not beautiful, compared to beautiful (Beautiful = 30.00%, Not Beautiful = 70.00%), to be reported as having shaped the participants’ musical taste (I-“Musical Taste and Identity”-B19, I-“Musical Taste and Identity”-NB19) and identity (I-“Musical Taste and Identity”-B20, I-“Musical Taste and Identity”-NB20). ”

Comment #14: Table 1. Examples of Responses from Feature Questions.p.l8-20: Regarding Table 1, why were so many negative responses regarding music that is not beautiful obtained? The question asks about music that is “often listened to but not beautiful.” These responses appear to be impressions of music that the respondents do not like.

Why do respondents “often listen to” songs with such negative characteristics?

Do these responses indicate a gap between beauty and preference? Or did the participants list songs that are “generally considered beautiful”?

Response:

Thank you for this question. We want to stress that negative feelings here include feelings like agitation and restlessness which while not positive, might be aesthetically valuable to a listener. In that vein, the occurrence of negative feelings may not be so surprising.

Comment #14: Round 2

Please incorporate the content here into the body text.

***Thank you. We realise now that it is important to mention this as you suggest. It has been added in the main text;

Lines 872-877, p. 56

“Negative emotions (such as restless, agitation, angst, hauntedness, frustration) were also reported, primarily in response to not beautiful pieces. The fact that participants chose to listen to music that evoked such emotions suggests that these pieces may hold aesthetic value despite not being considered beautiful and that experiencing these negative emotions through music may function as an emotional outlet, as reported in previous studies [126, 127].”

***

Reviewer #2: Dear authors,

Thank you for your additional work on the manuscript entitled “Unpacking musical beauty: sound, emotion, and impact differences across expertise and personality" (Manuscript ID: PONE-D-25-24652). Overall, I believe the manuscript has improved considerably, and I do not have any major concerns at this stage.

However, I have a few minor suggestions that may help strengthen the final version.

1. Language use & style: The manuscript is quite lengthy, and some sections are repetitive. For example, the sentence “In addition to lacking a clear definition, also frequently debated is how beauty should be conceptualised.” repeats the same idea. Similarly, in this passage the word "recognized" is repeated in consecutive sentences: "Here, it is recognised that it may be a positive experience for the listener when a piece of music unfolds as they anticipated [40,41]. However, it is also increasingly recognised that a degree of unexpectedness, surprise, or novelty in music can lead to greater feelings of pleasure.” These are just a couple of examples, but there are other similar instances throughout the text. I recommend carefully reading through the text with an eye toward tightening the language and reducing redundancy. You might also consider having a native English speaker review the manuscript to improve clarity and flow.

***Thank you for this comment. We have now carefully read and edited the manuscript. We hope the revised version has improved readability.

2. Long-term influences: The introduction mentions the long-term impact of experiencing beauty in music as a novel feature of this study. However, this aspect is not revisited in depth later in the manuscript. It is also unclear from the questionnaire whether participants were prompted to reflect on short-term or long-term effects. To clarify, it would be helpful to state in the introduction that the study explores both immediate and potentially lasting impacts. Consider referencing this again in the results and/or discussion.

*** Thank you for this. The questionnaire did not specify whether participants should reflect on short- or long-term impact. It was during the coding process we realised that it may be useful to separate these out. We agree with that the reviewer’s suggestion to be more consistent across the three sections of the manuscript. Therefore, we now, as advised, mention in the introduction that the impacts participants' report may be expected to range from shorter- or longer-term impacts and also, in the discussion, we emphasise that one might want to consider the impact of beauty in this nuanced way.

Lines 192-194, p.9: The Impact of experiencing beauty in music

“However, while it seems plausible that these shorter- and longer-term impacts may emerge from experiences of musical beauty specifically, this has not been examined in the extant literature on musical beauty.”

Lines 253-256, p. 11: The present study

“.., this study explored how the experience of musical beauty can be understood in terms of.., and the impact of musical beauty, including its potential short- and long-term effects.”

Lines 338-341, p. 17: Procedure

“Although this study aims to explore both short- and long-term impacts of experiencing musical beauty, participants were not instructed to report impacts as short- or long-term since it is difficult to specify these systematically.”

Lines 932-934, p.59: Discussion

“In line with the well-documented finding .., many participants reported what could be considered short-term impacts, noting that listening to both beautiful and not beautiful pieces..”

Lines 965-967, p. 60

“The potential difficulty in distinguishing oneself from others through beautiful music may explain why the affordance of a longer-term identity is instead obtained from the other music people value and often choose to listen to.”

3. Discussion: I felt that the new discussion section headings were too long and complex. To improve readability and structure, I suggest simplifying them. For example: “Intrinsic, structural, and aesthetic elements of beautiful music”, “Emotional responses to beautiful music”, “Short- and long-term impacts of beautiful music”, “Individual differences in experiences of beauty”

*** Thank you for your suggestion. We have simplified the headings as following.

- Beautiful music is evaluated in terms of its intrinsic, structural and aesthetic elements

-> The intrinsic, structural, and aesthetic determinants of beautiful music

- Beautiful music elicits calm, sadness, being moved and support

-> The characteristic emotional profile of beautiful music

- Beautiful music impacts mood, musical motivation, but mot identity

-> The scope and the nature of beautiful music impacts

- Individual differences in musical beauty experience

-> Individual differences in experiences of musical beauty

4. Figure 8 & Abstract: For Figure 8, consider providing a more balanced level of detail across all three sections. In the section "What beautiful music makes people feel" you list specific emotional responses reported by participants. However, in "What makes music beautiful", you only list broader categories (e.g., timbre, melody) without providing concrete examples of what was perceived as beautiful. Similarily, the abstract mentions emotional and impact-related findings but does not touch on the specific musical features that participants associated with beauty. Since many readers may only read the abstract or jump directly to the discussion and this nice figure, clearly and concisely summarizing all key findings in both Figure 8 and the abstract would strengthen the accessibility and impact of your work.

*** Thank you for your helpful suggestion. We have added the more details of musical features in both Figure 8 and the abstract.

Fig 8. Page 53

Lines 32-36, p. 2: Abstract

“Thematic analysis of text responses showed that participants consider intrinsic features of the music (mellow timbre, pleasing melodies and harmonies, slow tempo), cognitive factors (degree o

---

## [Decision Letter · Decision Letter 2]

20 Oct 2025

Unpacking musical beauty: sound, emotion, and impact differences across expertise and personality

PONE-D-25-24652R2

Dear Dr. Arthurs,

We’re pleased to inform you that your manuscript has been judged scientifically suitable for publication and will be formally accepted for publication once it meets all outstanding technical requirements.

Kind regards,

Andrea Schiavio

Academic Editor

PLOS ONE

Additional Editor Comments (optional):

Reviewers' comments:

Reviewer's Responses to Questions

**Comments to the Author**

1. If the authors have adequately addressed your comments raised in a previous round of review and you feel that this manuscript is now acceptable for publication, you may indicate that here to bypass the “Comments to the Author” section, enter your conflict of interest statement in the “Confidential to Editor” section, and submit your "Accept" recommendation.

Reviewer #1: All comments have been addressed

Reviewer #2: All comments have been addressed

2. Is the manuscript technically sound, and do the data support the conclusions?

Reviewer #1: Yes

Reviewer #2: Yes

3. Has the statistical analysis been performed appropriately and rigorously? 

Reviewer #1: Yes

Reviewer #2: Yes

4. Have the authors made all data underlying the findings in their manuscript fully available?

Reviewer #1: Yes

Reviewer #2: No

5. Is the manuscript presented in an intelligible fashion and written in standard English?

Reviewer #1: Yes

Reviewer #2: Yes

6. Review Comments to the Author

Reviewer #1: (No Response)

Reviewer #2: Dear authors,

Thank you for revising the article and addressing all my main comments. Overall, I feel that the article has improved in terms of readability and clarity thanks to your efforts to streamline it. The abstract and summary figure are also now more informative and will provide readers with a useful overview.

One minor point: I noticed that you intend to make the data available on OSF once the manuscript has been accepted. Please remember to include the OSF link in the final version of the manuscript.

Once this has been done, I recommend accepting the article for publication.

7. PLOS authors have the option to publish the peer review history of their article (what does this mean? ). If published, this will include your full peer review and any attached files.

**Do you want your identity to be public for this peer review?** For information about this choice, including consent withdrawal, please see our Privacy Policy .

Reviewer #1: No

Reviewer #2: **Yes: ** Julia Vigl

---

## [Editor Report · Acceptance letter]

PONE-D-25-24652R2

PLOS ONE

Dear Dr. Arthurs,

I'm pleased to inform you that your manuscript has been deemed suitable for publication in PLOS ONE. Congratulations! Your manuscript is now being handed over to our production team.

Kind regards,

on behalf of

Dr Andrea Schiavio

Academic Editor

PLOS ONE